# Multi-agent Performative Prediction with Greedy Deployment and Consensus Seeking Agents

**Qiang Li**      **Chung-Yiu Yau**      **Hoi-To Wai**

Department of Systems Engineering and Engineering Management
The Chinese University of Hong Kong, Shatin, Hong Kong SAR of China
`{liqiang, cyyau, htwai}@se.cuhk.edu.hk`

## Abstract

We consider a scenario where multiple agents are learning a common decision vector from data which can be influenced by the agents' decisions. This leads to the problem of multi-agent performative prediction (Multi-PfD). In this paper, we formulate Multi-PfD as a decentralized optimization problem that minimizes a sum of loss functions, where each loss function is based on a distribution influenced by the local decision vector. We first prove the necessary and sufficient condition for the Multi-PfD problem to admit a unique multi-agent performative stable (Multi-PS) solution. We show that enforcing consensus leads to a laxer condition for existence of Multi-PS solution with respect to the distributions' sensitivities, compared to the single agent case. Then, we study a decentralized extension to the greedy deployment scheme [Mendler-Dünner et al., 2020], called the `DSGD-GD` scheme. We show that `DSGD-GD` converges to the Multi-PS solution and analyze its non-asymptotic convergence rate. Numerical results validate our analysis.

## 1 Introduction

Traditional learning/prediction problems are often formulated with the assumption that data follows a *static* distribution, from which a decision vector is sought by solving a corresponding optimization problem. While this assumption holds for 'stationary' tasks such as image classification, many real world tasks are dynamical, involving data that could be influenced by decisions. In the latter case, the predictions are said to be *performative*. Example scenarios include when users are *strategic* [Hardt et al., 2016, Dong et al., 2018, Kleinberg and Raghavan, 2020] such as in training an E-mail spam classifier, where users (including spammers) can adapt to the classifier's rules to evade detection.

A common way to model *performative prediction* is via incorporating decision-dependent distribution in formulating the learning problem [Quiñonero-Candela et al., 2008]. Since the pioneering work by Perdomo et al. [2020], there is a growing literature in analyzing the performative prediction problem, which studied the convergence of learning algorithms to stable point of performative prediction [Mendler-Dünner et al., 2020, Drusvyatskiy and Xiao, 2020, Brown et al., 2022, Li and Wai, 2022, Wood et al., 2021], algorithms to find stationary solution of performative risk [Izzo et al., 2021, 2022, Miller et al., 2021, Ray et al., 2022], update timescales of agent [Zrnic et al., 2021], etc.

Most of the existing works are focused on the single agent (learner) setting. In this paper, we concentrate on the multi-agent performative prediction (Multi-PfD) problem. Here, the agents are *consensus-seeking* who seek a common decision vector that minimizes the sum of loss functions through communications on a graph/network. Our setup pertains to the wide-spread applications of consensus-type decentralized algorithms to machine learning which leverages the large amount of data at cooperating agents, such as clinical data [Warnat-Herresthal et al., 2021], financial data [Li and Wang, 2019], etc. In specific, adopting a consensus design can bring benefits such as better out-of-

sample (i.e., generalization) performance, the ability to utilize data observed at all agents, improved robustness to agent failure. As pointed out by [Perdomo et al., 2020], real world applications may involve user data that can be influenced by the optimization solution leading to distribution shifts. For example, in clinical data, the agents wish to train a model for treating some medical conditions. Upon the deployment of a model, the future patients may react to the just deployed model by overstating their symptoms accordingly, which increase their chances of receiving better treatment at the hospital. Another example is the training of an E-mail spam classifier, where each agent represents a regional server providing services to a subset of users. These users are in general different and may react to their serving agents differently, leading to heterogeneous and decision dependent data distributions. Notably, our setup differs from the recent works in [Narang et al., 2022, Piliouras and Yu, 2022] which consider a game theoretical setup of Multi-PfD; see Fig. 1 and §2.

For the algorithmic model, we study a decentralized extension of the greedy deployment scheme [Mendler-Dünner et al., 2020], particularly we concentrate on the decentralized stochastic gradient (DSGD) [Lian et al., 2017] algorithm with greedy deployment (`DSGD-GD`). Overall, the `DSGD-GD` scheme emulates a scenario where agents apply DSGD, a standard decentralized learning algorithm, while being *agnostic* to the performative nature of the prediction problem. In particular, upon each iteration, the non-converged local decisions will be deployed directly. Then, agents optimize their decision using stochastic gradient constructed from data with distribution shifted by local decision.

Tackling Multi-PfD with the `DSGD-GD` scheme yields a practical scenario of performative prediction in the multi-agent setting. We inquire the open questions: *When will the Multi-PfD problem admit a stable and consensual solution? If so, how fast does it take for* `DSGD-GD` *to converge to such solution? Does the limited communication in* `DSGD-GD` *impair its convergence rate?* We provide affirmative answers to the above. Our idea hinges on adopting the concept of *performative stable* solution [Perdomo et al., 2020], which is a fixed point solution resulted from the interplay between agent(s) and data that react to the agent(s)' decisions. In particular, we focus on analyzing the multi-agent performative stable (Multi-PS) solution in our problem.

To our best knowledge, this paper provides the first study and analysis of Multi-PfD with consensus seeking agents via a practical `DSGD-GD` scheme. We highlight the following key contributions:

- We provide a *necessary and sufficient* condition on the sensitivity of decision dependent data distributions for the existence and uniqueness of the Multi-PS solution. An interesting finding is that the Multi-PS solution exists even if some of the performative prediction problems of individual agents are unstable. The consensus seeking behavior in Multi-PfD tends to *stabilize* the problem.

- We study the `DSGD-GD` scheme and analyze its convergence towards the Multi-PS solution. We first show that the scheme is convergent under the same sufficient condition for existence of Multi-PS solution. With an appropriate step size rule, in expectation, the squared distance between the Multi-PS solution and the iterates decays as $\mathcal{O}(1/t)$, and the squared consensus distance decays as $\mathcal{O}(1/t^2)$, where $t$ is the iteration number. Our fine-grained analysis also reveals that heterogeneous users, poorly connected graph may slow down the convergence of `DSGD-GD`, but only so in the transient, and the asymptotic rate has a linear speedup property.

- To validate our analysis, we conduct numerical experiments on synthetic data and real data. Particularly, we consider the Gaussian mean estimation on synthetic data to validate our theory, and the logistics regression problem on the `spambase` dataset.

The paper is organized as follows. §2 introduces the Multi-PfD problem and `DSGD-GD` scheme. §3 presents the theoretical results on Multi-PfD, `DSGD-GD` together with a proof outline. Finally, §4 shows numerical experiments to validate our analysis. All proofs can be found in the appendix.

**Related Works.** As mentioned, the main ingredient of `DSGD-GD` scheme is the classical DSGD algorithm. The latter was introduced in [Ram et al., 2010, Bianchi and Jakubowicz, 2012, Sayed, 2014] and is recently popularized for decentralized learning. Of relevance to our work are [Lian et al., 2017] which demonstrated a linear speed up against centralized SGD, and [Pu et al., 2021] with a refined analysis, also see [Kong et al., 2021, Bars et al., 2022] on the effects of consensus steps and topology. We also mention recent advances to improve the efficiencies of DSGD in [Tang et al., 2018, Lan et al., 2020, Koloskova et al., 2019]. Our analysis entails fresh challenges as the sought Multi-PS solution cannot be described explicitly as an optimal solution to the Multi-PfD problem.

Another line of relevant works pertains to (multi-agent) reinforcement learning whose formulation also entails a decision-dependent distribution. To this end, policy gradient algorithms are analyzed in

[Zhang et al., 2020, Karimi et al., 2019], and they have been recently extended to the multi-agent setting [Zhang et al., 2018, Chen et al., 2021]; also see the survey [Zhang et al., 2021]. Most of the above works showed convergence to a stationary point which may not be unique. This is in contrast to our analysis of DSGD-GD which converges to the unique Multi-PS solution.

## 2    Problem Setup

Consider a scenario where there are $n$ agents connected on an undirected and connected graph $G = (V, E)$ such that $V = \{1, \ldots, n\}$, $E \subseteq V \times V$. Note that we include self-loops such that $(i, i) \in E$ for any $i \in V$. Each agent $i \in V$ draws samples from the $i$th population of users. The latter is characterized by the distribution $\mathcal{D}_i(\boldsymbol{\theta}_i)$ supported on $\mathsf{Z} \subseteq \mathbb{R}^p$, $p \in \mathbb{N}$ and parameterized by the agent's decision vector $\boldsymbol{\theta}_i \in \mathbb{R}^d$, $d \in \mathbb{N}$. In other words, the $i$th population of users are only influenced by the $i$th agent's decision vector. Note that the populations of users can be *heterogeneous* such that $\mathcal{D}_i(\boldsymbol{\theta}) \neq \mathcal{D}_j(\boldsymbol{\theta}')$, $i \neq j$, even if $\boldsymbol{\theta} = \boldsymbol{\theta}'$.

The agents aim to find a *common decision vector* $\boldsymbol{\theta} \in \mathbb{R}^d$ in a collaborative fashion that minimizes the average of local losses. Consider the *multi-agent performative prediction (Multi-PfD) problem*:

$$\min_{\boldsymbol{\theta}_i \in \mathbb{R}^d, \, i=1,\ldots,n} \; \tfrac{1}{n} \sum_{i=1}^n \mathbb{E}_{Z_i \sim \mathcal{D}_i(\boldsymbol{\theta}_i)} \big[ \ell(\boldsymbol{\theta}_i; Z_i) \big] \;\; \text{s.t.} \;\; \boldsymbol{\theta}_i = \boldsymbol{\theta}_j, \; \forall \, (i,j) \in E. \tag{1}$$

Since $G$ is connected, the constraint $\boldsymbol{\theta}_i = \boldsymbol{\theta}_j$ for $(i, j) \in E$ enforces the decision to be in *consensus* across the $n$ agents. In the above, $\ell(\boldsymbol{\theta}_i; Z_i)$ is the loss function of the fitness of the decision vector $\boldsymbol{\theta}_i$ with respect to (w.r.t.) the sample $Z_i$. The expected value $\mathbb{E}_{Z_i \sim \mathcal{D}_i(\boldsymbol{\theta}_i)}[\ell(\boldsymbol{\theta}_i; Z_i)]$ corresponds to the loss at agent $i$ w.r.t. the samples from the $i$th population of users.

**Example 1.** *We describe a strategic binary classification problem with linear utility for users to illustrate the application of* (1). *The sample $Z_i$ is defined by a tuple $Z_i = (\boldsymbol{X}_i, Y_i) \in \mathbb{R}^d \times \{0, 1\}$ of feature and binary label, and the loss function is taken as the logistic regression function:*

$$\ell(\boldsymbol{\theta}; Z_i) = \log \big( 1 + \exp(\langle \boldsymbol{X}_i \, | \, \boldsymbol{\theta} \rangle) \big) - Y \langle \boldsymbol{X}_i \, | \, \boldsymbol{\theta} \rangle + \tfrac{\beta}{2} \| \boldsymbol{\theta} \|^2, \tag{2}$$

*where $\beta > 0$ is a regularization parameter. The above loss function quantifies the mismatches between the classifier $\boldsymbol{\theta}$ and the given data tuple $Z_i = (\boldsymbol{X}_i, Y_i)$.*

*On the other hand, the distribution $\mathcal{D}_i(\boldsymbol{\theta}_i)$ is controlled by the $i$th population of users. The distribution is* decision dependent *such that the sample $Z_i \sim \mathcal{D}_i(\boldsymbol{\theta}_i)$ depends on the $i$th decision $\boldsymbol{\theta}_i$. Observing the $i$th decision $\boldsymbol{\theta}_i$, users of the $i$th population provides samples that are modified via a linear utility such that $Z_i = (\boldsymbol{X}_i, Y_i) \sim \mathcal{D}_i(\boldsymbol{\theta}_i)$ is given by*

$$\boldsymbol{X}_i = \arg\max_{\hat{\boldsymbol{X}} \in \mathbb{R}^d} \big\{ \langle \boldsymbol{\theta}_i \, | \, \hat{\boldsymbol{X}} \rangle - \tfrac{1}{2\epsilon_i} \| \hat{\boldsymbol{X}} - \boldsymbol{X} \|^2 \big\}, \; Y_i = Y \;\; \text{with} \;\; (\boldsymbol{X}, Y) \sim \mathcal{D}_i^\circ, \tag{3}$$

*for some $\epsilon_i > 0$, where $\mathcal{D}_i^\circ$ is a base data distribution of the $i$th population. Note that in the above, we have the closed form solution $\boldsymbol{X}_i = \boldsymbol{X} + \epsilon_i \boldsymbol{\theta}_i$. Tackling* (1) *leads to a common classifier $\overline{\boldsymbol{\theta}}$ which takes the effects of the heterogeneous decision dependent distributions into account.*

The Multi-PfD problem (1) comprises of a stochastic objective function with a decision dependent distribution. In particular, for each agent $i$, the distribution $\mathcal{D}_i(\boldsymbol{\theta}_i)$ captures a feedback mechanism where users of the $i$th population react to the decision made by the $i$th agent. See Fig. 1 (left) for an illustration. Due to non-convexity, the performative risk in (1) can be difficult to minimize. In this paper, we are interested in the *multi-agent performative stable* (Multi-PS) solution:

$$\boldsymbol{\theta}^{PS} = \mathcal{M}(\boldsymbol{\theta}^{PS}) := \arg\min_{\boldsymbol{\theta} \in \mathbb{R}^d} \tfrac{1}{n} \sum_{i=1}^n \mathbb{E}_{Z_i \sim \mathcal{D}_i(\boldsymbol{\theta}^{PS})}[\ell(\boldsymbol{\theta}; Z_i)] \tag{4}$$

which approximately solves (1). Notice that $\boldsymbol{\theta}^{PS}$ is defined to be a fixed point of the map $\mathcal{M} : \mathbb{R}^d \to \mathbb{R}^d$. The existence and uniqueness of $\boldsymbol{\theta}^{PS}$ will be shown under mild condition in Proposition 1.

**Comparison to Existing Works.** Our setup differs from recent works in [Piliouras and Yu, 2022, Narang et al., 2022]. Unlike ours, both works considered a game theoretical setting where the agents do not directly communicate their decisions. Instead, agents are coupled through the data distributions which depend on the global decision profile $\vartheta := (\boldsymbol{\theta}_1, \ldots, \boldsymbol{\theta}_n)$.

In [Narang et al., 2022], agent/player $i$ seeks to solve $\min_{\boldsymbol{\theta}_i} \mathbb{E}_{Z_i \sim \mathcal{D}_i(\vartheta)}[\ell_i(\vartheta; Z_i)]$ where the local distribution $\mathcal{D}_i$ depends on the global decision $\vartheta$ [Fig. 1 (middle)]; while Piliouras and Yu [2022] study a slightly different setting where the goal of agent/player $i$ is to solve $\min_{\boldsymbol{\theta}_i} \mathbb{E}_{Z_i \sim \mathcal{D}(\vartheta)}[\ell(\boldsymbol{\theta}_i; Z_i)]$

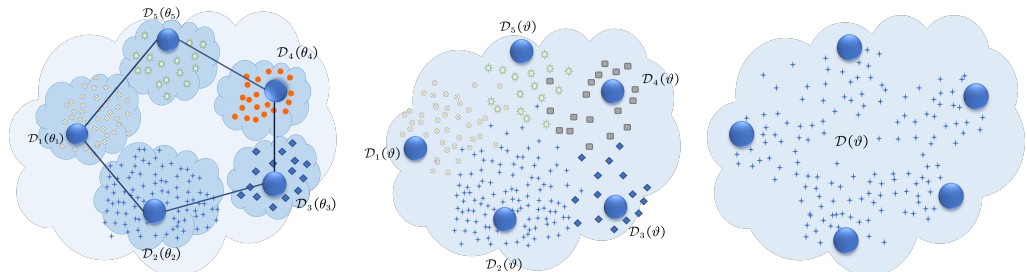

Figure 1: **Comparing the frameworks of multi-agent/player performative prediction**. (Left) This work: agent deploys $\boldsymbol{\theta}_i$ and local distribution $\mathcal{D}_i(\boldsymbol{\theta}_i)$ is affected by $\boldsymbol{\theta}_i$ only; the agents communicate while aiming to reach consensus $\boldsymbol{\theta}_1 = \cdots = \boldsymbol{\theta}_n$ eventually. (Middle) Narang et al. [2022]: agent deploys $\boldsymbol{\theta}_i$ but local distribution is affected by joint decision $\mathcal{D}_i(\boldsymbol{\theta}_1, \ldots, \boldsymbol{\theta}_n)$. (Right) Piliouras and Yu [2022]: similar to [Narang et al., 2022] but the distribution is global $\mathcal{D}(\boldsymbol{\theta}_1, \ldots, \boldsymbol{\theta}_n)$.

[Fig. 1 (right)]. Both works showed convergence to a stable equilibria via (stochastic) gradient-type updates. In contrast, our work considers cooperative agents who are willing to share their decisions with other agents. Under this setting, we show that a natural decentralized scheme achieves convergence to a consensual decision that is performatively stable.

**Decentralized Scheme.** To find an approximate solution to (1) in a cooperative fashion, the natural approach is to deploy a decentralized scheme where agents communicate limited amount of information with neighbors that are directly connected on $G$. Furthermore, we focus on a scenario where agents are *agnostic* to that their users may react to the agents' decisions. The above motivates us to study the following variant of the standard DSGD algorithm [Lian et al., 2017].

Let us define a mixing matrix $\boldsymbol{W} \in \mathbb{R}_+^{n \times n}$ on $G$ such that $W_{ij} = W_{ji} > 0$ if $(i, j) \in E$, otherwise $W_{ij} = 0$ if $(i, j) \notin E$, satisfying $\sum_{i=1}^n W_{ij} = 1$ for any $i = 1, \ldots, n$. We consider the scheme:

---

**DSGD with Greedy Deployment (DSGD-GD) Scheme**

At iteration $t = 0, 1, \ldots$, for any $i \in V$, agent $i$ updates his/her decision $(\boldsymbol{\theta}_i^t)$ by the recursion consisting of two phases

$$\text{(Phase 1) } Z_i^{t+1} \sim \mathcal{D}_i(\boldsymbol{\theta}_i^t) \quad \Big| \quad \text{(Phase 2) } \boldsymbol{\theta}_i^{t+1} = \sum_{j=1}^n W_{ij} \boldsymbol{\theta}_j^t - \gamma_{t+1} \nabla \ell(\boldsymbol{\theta}_i^t; Z_i^{t+1}), \tag{5}$$

where $\gamma_{t+1} > 0$ is a step size. Note that $\nabla \ell(\boldsymbol{\theta}_i^t; Z_i^{t+1})$ denotes the gradient taken w.r.t. the first argument $\boldsymbol{\theta}_i^t$, and the samples $Z_i^{t+1}$ at each agent are independent of each other.

---

The above combines the DSGD algorithm [Lian et al., 2017] with greedy deployment [Mendler-Dünner et al., 2020]. For iteration $t \geq 0$, the scheme can be described with two phases:

In Phase 1, agent $i$ deploys $\boldsymbol{\theta}_i^t$ at the $i$th population, whose users react to the decision and reveal a sample $Z_i^{t+1} \sim \mathcal{D}_i(\boldsymbol{\theta}_i^t)$ to the agent. Note that this is a greedy deployment scheme since the deployed decision $\boldsymbol{\theta}_i^t$ may not be stable w.r.t. (1). In Phase 2, agent $i$ receives the current decisions $\boldsymbol{\theta}_j^t$ from his/her neighbors $j \in \mathcal{N}_i$ and update $\boldsymbol{\theta}_i^{t+1}$ according to the 'consensus' + 'stochastic gradient' step. Here, the 'stochastic gradient' step is based on the local decision $\boldsymbol{\theta}_i^t$ and sample $Z_i^{t+1}$ taken in phase one with greedy deployment. We remark that throughout the DSGD-GD scheme, the agents remain unaware of the performative behavior of their users.

Notice that the DSGD-GD scheme can be interpreted as a DSGD algorithm deploying *biased* stochastic gradient updates. Denote $\mathbb{E}_t[\cdot]$ as the conditional expectation up to iteration $t$, we observe

$$\mathbb{E}_t[\nabla_{\boldsymbol{\theta}_i^t} \ell(\boldsymbol{\theta}_i^t; Z_i^{t+1})] = \mathbb{E}_{Z_i \sim \mathcal{D}_i(\boldsymbol{\theta}_i^t)}[\nabla_{\boldsymbol{\theta}_i^t} \ell(\boldsymbol{\theta}_i^t; Z_i)] \neq \nabla_{\boldsymbol{\theta}_i^t} \big\{ \mathbb{E}_{Z_i \sim \mathcal{D}_i(\boldsymbol{\theta}_i^t)}[\ell(\boldsymbol{\theta}_i^t; Z_i)] \big\}. \tag{6}$$

The unbiased gradient on the r.h.s. also involves derivative w.r.t. the decision in the distribution. Analyzing the convergence of DSGD-GD therefore requires different techniques.

# 3  Main Results

This section studies the convergence of the DSGD-GD scheme and demonstrates that the latter can approximately solve the Multi-PfD problem (1). To facilitate our discussions, we first define:

$$f_i(\boldsymbol{\theta}; \overline{\boldsymbol{\theta}}) := \mathbb{E}_{Z_i \sim \mathcal{D}_i(\overline{\boldsymbol{\theta}})}[\ell(\boldsymbol{\theta}; Z_i)], \ \ f(\boldsymbol{\theta}; \overline{\boldsymbol{\theta}}) := \tfrac{1}{n}\textstyle\sum_{i=1}^n f_i(\boldsymbol{\theta}; \overline{\boldsymbol{\theta}}). \tag{7}$$

Note that the first arguments in $f_i, f$ denote the agent's decision and the second argument is the deployed decision known to the population of users. For the rest of this paper, unless otherwise specified, $\nabla \ell(\boldsymbol{\theta}; Z), \nabla f_i(\boldsymbol{\theta}; \boldsymbol{\theta}'), \nabla f(\boldsymbol{\theta}; \boldsymbol{\theta}')$ denote the gradients taken w.r.t. the first argument $\boldsymbol{\theta}$.

Using the above notations, (1) is equivalent to $\min_{\boldsymbol{\theta}} f(\boldsymbol{\theta}; \boldsymbol{\theta})$. We consider the set of assumptions:

**A1.** *Fix any $\bar{\boldsymbol{\theta}} \in \mathbb{R}^d$, the function $f(\boldsymbol{\theta}; \bar{\boldsymbol{\theta}})$ is $\mu$-strongly convex in $\boldsymbol{\theta}$ such that*

$$f(\boldsymbol{\theta}'; \bar{\boldsymbol{\theta}}) \geq f(\boldsymbol{\theta}; \bar{\boldsymbol{\theta}}) + \langle \nabla f(\boldsymbol{\theta}; \bar{\boldsymbol{\theta}}) \,|\, \boldsymbol{\theta}' - \boldsymbol{\theta} \rangle + (\mu/2)\|\boldsymbol{\theta}' - \boldsymbol{\theta}\|^2, \ \forall \, \boldsymbol{\theta}', \boldsymbol{\theta} \in \mathbb{R}^d. \tag{8}$$

**A2.** *For any $i = 1, \ldots, n$, the loss function $\ell(\boldsymbol{\theta}; z)$ is $L$-smooth such that*

$$\|\nabla \ell(\boldsymbol{\theta}; z) - \nabla \ell(\boldsymbol{\theta}'; z')\| \leq L\{\|\boldsymbol{\theta} - \boldsymbol{\theta}'\| + \|z - z'\|\}, \ \forall \, \boldsymbol{\theta}', \boldsymbol{\theta} \in \mathbb{R}^d, z, z' \in \mathsf{Z}. \tag{9}$$

**A3.** *For any $i = 1, \ldots, n$, there exists a constant $\epsilon_i > 0$ such that*

$$\mathcal{W}_1(\mathcal{D}_i(\boldsymbol{\theta}), \mathcal{D}_i(\boldsymbol{\theta}')) \leq \epsilon_i \|\boldsymbol{\theta} - \boldsymbol{\theta}'\|, \ \forall \, \boldsymbol{\theta}', \boldsymbol{\theta} \in \mathbb{R}^d, \tag{10}$$

*where $\mathcal{W}_1(\mathcal{D}, \mathcal{D}')$ denotes the Wasserstein-1 distance between the distributions $\mathcal{D}, \mathcal{D}'$.*

A1, A2 require the loss functions to be strongly convex and smooth, while A3 states that the amount of distribution shift caused by the reaction of $i$th population to the agent's decision grows linearly with difference in decision. These assumptions are standard in the literature, e.g., [Perdomo et al., 2020, Mendler-Dünner et al., 2020, Drusvyatskiy and Xiao, 2020]. To simplify notations, we define the average, maximum sensitivity as $\epsilon_{\mathsf{avg}} := \sum_{i=1}^n \epsilon_i/n, \epsilon_{\mathsf{max}} := \max_{i=1,\ldots,n} \epsilon_i$, respectively.

Our first result establishes the existence of the Multi-PS solution $\boldsymbol{\theta}^{PS}$ satisfying $\nabla f(\boldsymbol{\theta}^{PS}; \boldsymbol{\theta}^{PS}) = \mathbf{0}$:

**Proposition 1 (Existence and Uniqueness of $\boldsymbol{\theta}^{PS}$).** *Under A1–A3. Define the map $\mathcal{M} : \mathbb{R}^d \to \mathbb{R}^d$*

$$\mathcal{M}(\boldsymbol{\theta}) = \arg\min_{\boldsymbol{\theta}' \in \mathbb{R}^d} \tfrac{1}{n}\textstyle\sum_{i=1}^n f_i(\boldsymbol{\theta}'; \boldsymbol{\theta}) \tag{11}$$

*If $\epsilon_{\mathsf{avg}} < \mu/L$, then the map $\mathcal{M}(\boldsymbol{\theta})$ is a contraction with the unique fixed point $\boldsymbol{\theta}^{PS} = \mathcal{M}(\boldsymbol{\theta}^{PS})$. If $\epsilon_{\mathsf{avg}} \geq \mu/L$, then there exists an instance of (11) where $\lim_{T \to \infty} \|\mathcal{M}^T(\boldsymbol{\theta})\| = \infty$.*

See §A for the proof. Note the inverse condition number $\mu/L$ yields a *tight* threshold on the sensitivity of the population for the stability of Multi-PfD. Our result can be viewed as the multi-agent extension to [Perdomo et al., 2020, Prop. 4.1]. Notice that while $\boldsymbol{\theta}^{PS}$ may not solve (1), it yields an approximate solution to the latter, depending on the magnitude of $\epsilon_{\mathsf{avg}}$; see [Perdomo et al., 2020].

Our result shows that Multi-PfD has a relaxed requirement for the existence of Multi-PS solution as it only depends on the *average sensitivity* $\epsilon_{\mathsf{avg}}$. Consider when $\epsilon_i$ exceeds $\mu/L$ for the population served by $i$th agent, now there may not exist a performative stable solution for the *individual* agent [Perdomo et al., 2020]. Meanwhile, by Proposition 1, the *Multi-PS* solution still exists as long as $\epsilon_{\mathsf{avg}} < \mu/L$, e.g., when there are agents in the network with less sensitive users.

We also compare Proposition 1 to [Narang et al., 2022]. Note that the latter considers decision-dependent distributions $\mathcal{D}_i(\vartheta)$ with $\vartheta := (\boldsymbol{\theta}_1, \ldots, \boldsymbol{\theta}_n)$ involving all decisions, which is different from our setting. Nevertheless, under similar conditions to A1–A3, Narang et al. [2022, Theorem 1] shows the performative stable equilibria exists if $\sum_{i=1}^n \epsilon_i^2 < \mu^2/L^2$. When $\epsilon_i = \epsilon_{\mathsf{avg}}$, it yields $\epsilon_{\mathsf{avg}} < \mu/(\sqrt{n}L)$ which is more restrictive than the requirement in Proposition 1.

The relaxed condition in Proposition 1 can be attributed to the consensus seeking nature of (1), where agents with less sensitive users serve as mediators that prevent divergence of the map (11).

**Convergence Analysis of DSGD-GD Scheme.** Next, we focus on analyzing the convergence of the DSGD-GD scheme. We require the following assumptions:

**A4.** *The non-negative matrix $\boldsymbol{W}$ is doubly stochastic, i.e., $\boldsymbol{W}\mathbf{1} = \boldsymbol{W}^\top \mathbf{1} = \mathbf{1}$. There exists a constant $\rho \in (0, 1]$ such that $\|\boldsymbol{W} - (1/n)\mathbf{1}\mathbf{1}^\top\|_2 \leq 1 - \rho$.*

**A5.** *For any $i = 1, \ldots, n$ and fixed $\boldsymbol{\theta} \in \mathbb{R}^d$, there exists $\sigma \geq 0$ such that*

$$\mathbb{E}_{Z_i \sim \mathcal{D}_i(\boldsymbol{\theta})}[\|\nabla \ell(\boldsymbol{\theta}; Z_i) - \nabla f_i(\boldsymbol{\theta}; \boldsymbol{\theta})\|^2] \leq \sigma^2 (1 + \|\boldsymbol{\theta} - \boldsymbol{\theta}^{PS}\|^2). \tag{12}$$

A4 is a standard assumption on the mixing matrix, for example if $G$ is a connected graph, then $\boldsymbol{W}$ satisfying the condition can be constructed [Boyd et al., 2004]. Meanwhile, A5 bounds the variance of the 'stochastic gradient' $\nabla \ell(\boldsymbol{\theta}; Z_i)$ as the expected value to the latter yields $\mathbb{E}_{Z_i \sim \mathcal{D}_i(\boldsymbol{\theta})}[\nabla \ell(\boldsymbol{\theta}; Z_i)] = \nabla f(\boldsymbol{\theta}; \boldsymbol{\theta})$, where the gradients are taken w.r.t. the first argument $\boldsymbol{\theta}$. Moreover, we assume that:

**A6.** *For any $i = 1, \ldots, n$, there exists $\varsigma \geq 0$ such that*

$$\|\nabla f(\boldsymbol{\theta}; \boldsymbol{\theta}) - \nabla f_i(\boldsymbol{\theta}; \boldsymbol{\theta})\|^2 \leq \varsigma^2 (1 + \|\boldsymbol{\theta} - \boldsymbol{\theta}^{PS}\|^2), \ \forall \ \boldsymbol{\theta} \in \mathbb{R}^d. \tag{13}$$

It bounds the *heterogeneity* of the locally observed samples. As $\nabla f_i(\boldsymbol{\theta}; \boldsymbol{\theta}) = \mathbb{E}_{Z_i \sim \mathcal{D}_i(\boldsymbol{\theta})}[\nabla \ell(\boldsymbol{\theta}; Z_i)]$, if $\mathcal{D}_i(\boldsymbol{\theta}) = \mathcal{D}_j(\boldsymbol{\theta})$, the constant will be $\varsigma = 0$ when the population associated with each agent produces identically distributed samples with the same deployed decision vector. Otherwise, $\varsigma > 0$ measures degree of heterogeneity across populations. We remark that even when $\mathcal{D}_i(\boldsymbol{\theta}) = \mathcal{D}_j(\boldsymbol{\theta})$, the samples $Z_i^{t+1}, Z_j^{t+1}$ may still not be identically distributed *during the* `DSGD-GD` *iterations* since the decision vectors $\boldsymbol{\theta}_i^t, \boldsymbol{\theta}_j^t$ may not be in consensus when $t < \infty$.

A6 also implies $\max_{i=1,\ldots,n} \|\nabla f_i(\boldsymbol{\theta}^{PS}; \boldsymbol{\theta}^{PS})\|^2 \leq \varsigma^2$. In fact, as we show in (52) of the appendix, it suffices to prove the convergence of `DSGD-GD` without A6 as in [Pu et al., 2021, Yuan and Alghunaim, 2021]. We proceed our analysis with A6 to extract a tighter bound especially when $\varsigma$ is small.

In both A5, A6, we allowed the upper bounds to grow with $\mathcal{O}(1 + \|\boldsymbol{\theta} - \boldsymbol{\theta}^{PS}\|^2)$, e.g. A5 is similar to [Pu et al., 2021, Assumption 1]. This is a weaker condition than the commonly assumed uniform bound of $\mathcal{O}(1)$, e.g., in [Lian et al., 2017]. For example, it covers situations when the quadratic components in $\ell(\cdot)$ depends on both $Z$ and $\boldsymbol{\theta}$. Note that this relaxation implies that the variance/heterogeneity can be unbounded, leading to additional challenges in our analysis.

Define the following quantities and constants: fix any $\delta > 0$,

$$\overline{\boldsymbol{\theta}}^t := (1/n) \sum_{i=1}^n \boldsymbol{\theta}_i^t, \ \ \widetilde{\boldsymbol{\theta}}^t := \overline{\boldsymbol{\theta}}^t - \boldsymbol{\theta}^{PS}, \ \ \boldsymbol{\Theta}_o^t := (\boldsymbol{\theta}_1^t \ \cdots \ \boldsymbol{\theta}_n^t) - \overline{\boldsymbol{\theta}}^t \mathbf{1}^\top, \ \ \widetilde{\mu} := \mu - (1+\delta)\epsilon_{\mathsf{avg}} L,$$

$$c_1 := \frac{L(1 + \epsilon_{\mathsf{max}})^2}{2n\delta\epsilon_{\mathsf{avg}}}, \ \ c_2 := 4 \left( \frac{\sigma^2}{n} + L^2(1 + \epsilon_{\mathsf{max}})^2 \right), \ \ c_3 := 12\sigma^2 + 18L^2(1 + \epsilon_{\mathsf{max}})^2. \tag{14}$$

Note that $\widetilde{\boldsymbol{\theta}}^t$ is the distance between the $t$th averaged iterate $\overline{\boldsymbol{\theta}}^t$ and $\boldsymbol{\theta}^{PS}$, while $\boldsymbol{\Theta}_o^t$ is a $d \times n$ matrix of the $t$th *consensus error*. The following theorem establishes the convergence rate of `DSGD-GD`:

---

**Theorem 1.** *Under A1–A6 and the condition on average sensitivity that $\epsilon_{\mathsf{avg}} < \frac{\mu}{(1+\delta)L}$. Suppose the step sizes $\{\gamma_t\}_{t \geq 1}$ satisfy $\gamma_{t+1} \leq \gamma_t$ for any $t \geq 1$,*

$$\sup_{t \geq 1} \gamma_t \leq \min \left\{ \frac{4}{\widetilde{\mu}}, \ \frac{\widetilde{\mu}}{c_2}, \ \frac{\rho}{\sqrt{2c_3}}, \ \sqrt{\frac{\rho^2 \widetilde{\mu}}{192c_1(\sigma^2 + \varsigma^2)}}, \ \frac{\rho c_1}{4\widetilde{\mu}c_1 + \rho c_2} \right\}, \tag{15}$$

*and the condition $\frac{\gamma_t}{\gamma_{t+1}} \leq \min\{\sqrt{1 + (\widetilde{\mu}/4)\gamma_{t+1}^2}, \sqrt[3]{1 + (\widetilde{\mu}/4)\gamma_{t+1}^3}, 1 + \rho/(4 - 2\rho)\}$ for any $t \geq 1$. Then, the iterates generated by `DSGD-GD` admit the following bound for any $t \geq 0$,*

$$\mathbb{E}\left[\left\|\widetilde{\boldsymbol{\theta}}^{t+1}\right\|^2\right] \leq \prod_{i=1}^{t+1} \left(1 - \frac{\widetilde{\mu}\gamma_i}{2}\right) \mathsf{D} + \frac{288c_1(\sigma^2 + \varsigma^2)}{\rho^2 \widetilde{\mu}} \gamma_{t+1}^2 + \frac{8\sigma^2}{\widetilde{\mu}n} \gamma_{t+1}, \tag{16}$$

$$\mathbb{E}\left[\frac{1}{n}\left\|\boldsymbol{\Theta}_o^{t+1}\right\|_F^2\right] \leq \left(1 - \frac{\rho}{2}\right)^{t+1} \frac{1}{n}\left\|\boldsymbol{\Theta}_o^0\right\|_F^2 + \frac{2(9 + 12\overline{\Delta})(\sigma^2 + \varsigma^2)}{\rho^2} \gamma_{t+1}^2, \tag{17}$$

*where $\mathsf{D} := \|\widetilde{\boldsymbol{\theta}}^0\|^2 + \gamma_1 \frac{8c_1}{n\rho} \|\boldsymbol{\Theta}_o^0\|^2$ denotes the initial error, $\overline{\Delta} := \mathsf{D} + \frac{3}{2} + \frac{8\sigma^2}{c_2 n}$, and we recall the definitions of the constants $c_1, c_2, c_3$ from (14).*

---

The free parameter $\delta > 0$ in (14) can be chosen arbitrarily. Thus, according to Theorem 1, `DSGD-GD` converges when the average sensitivity $\epsilon_{\mathsf{avg}}$ is strictly below the threshold $\mu/L$ in Proposition 1, i.e.,

the same sufficient and necessary condition that guarantees the existence of $\boldsymbol{\theta}^{PS}$. Further, our result holds for general step size rules such as constant step size and diminishing step size.

To yield $\mathbb{E}[\|\widetilde{\boldsymbol{\theta}}^t\|^2] \to 0$, a common option is $\gamma_t = \frac{a_0}{a_1+t}$ for some $a_0, a_1 > 0$. In the latter case as $\gamma_t = \mathcal{O}(1/t)$, we observe from (16), (17) that the distance to performative stable solution $\|\widetilde{\boldsymbol{\theta}}^t\|^2$ converges to zero as $\mathcal{O}(1/t)$, while the consensus error $\|\boldsymbol{\Theta}_o^t\|_F^2$ converges to zero as $\mathcal{O}(1/t^2)$. We remark that our analysis also applies to the case of time varying graph; see §G for a proof sketch.

On the other hand, the bound (16) can be simplified to

$$\mathbb{E}[\|\overline{\boldsymbol{\theta}}^t - \boldsymbol{\theta}^{PS}\|^2] \lesssim \prod_{i=1}^t \left(1 - \frac{\widetilde{\mu}\gamma_i}{2}\right) + \frac{L(\sigma^2 + \varsigma^2)}{n\delta\widetilde{\mu}\rho^2\epsilon_{\mathsf{avg}}}\gamma_t^2 + \frac{\sigma^2}{n\widetilde{\mu}}\gamma_t. \tag{18}$$

As seen, the error is controlled by three terms. The first term is a transient term that consists of the product $\prod_{i=1}^t (1 - \widetilde{\mu}\gamma_i)$. It decays sub-geometrically and is scaled by the initial error D. The second term is a transient term and is affected by the spectral gap of mixing matrix $\rho$, the degree of heterogeneity $\varsigma$, etc. It decays as $\mathcal{O}(\gamma_t^2)$. Finally, the last term is a fluctuation term that only depends on the averaged noise variance $\mathcal{O}(\sigma^2/n)$. It decays at the slowest rate as $\mathcal{O}(\gamma_t)$.

**Effects of Network Topology and Heterogeneity.** An interesting observation from (18) is on how the network topology ($\rho$) and heterogeneity across local sample distribution ($\varsigma$) affects the convergence behavior of DSGD-GD. First, we note that the last term of $\mathcal{O}(\gamma_t \sigma^2/(\widetilde{\mu}n))$ is similar to the fluctuation term in SGD using the batch size of $n$ under a centralized setting, e.g., [Moulines and Bach, 2011]. Second, the constants $\rho, \varsigma$ only affect the term of rate $\mathcal{O}(\gamma_t^2)$.

In the case with diminishing step size $\gamma_t = \frac{a_0}{a_1+t}$, the second term of (18) will vanish at a faster rate than the last term. Particularly, if for some C > 0,

$$\gamma_t \leq \mathrm{C} \cdot \delta\rho^2\epsilon_{\mathsf{avg}}\sigma^2[L(\sigma^2 + \varsigma^2)]^{-1} \tag{19}$$

then (18) will be dominated by $\mathcal{O}(\gamma_t \sigma^2/(\widetilde{\mu}n))$. In other words, the effects of the network topology and heterogeneous population vanish asymptotically as $\overline{\boldsymbol{\theta}}^t \to \boldsymbol{\theta}^{PS}$.

As such, the 'linear speedup' behavior of DSGD in [Lian et al., 2017, Pu et al., 2021] can be extended to Multi-PfD with the DSGD-GD scheme. Again, we highlight that the consensus seeking behavior of agents has led to such speedup in convergence towards the Multi-PS solution.

## 3.1 Proof of Theorem 1

We outline the main steps in proving Theorem 1. As a preparatory step, we borrow the following lemma on the Lipschitzness of $\nabla f_i(\boldsymbol{\theta}, \boldsymbol{\theta})$ from [Drusvyatskiy and Xiao, 2020, Lemma 2.1]:

**Lemma 2 (Continuity of $\nabla f_i$).** *Under A2, A3. For any $\boldsymbol{\theta}_0, \boldsymbol{\theta}_1, \boldsymbol{\theta}, \boldsymbol{\theta}' \in \mathbb{R}^d$, it holds:*

$$\|\nabla f_i(\boldsymbol{\theta}_0; \boldsymbol{\theta}) - \nabla f_i(\boldsymbol{\theta}_1; \boldsymbol{\theta}')\| \leq L\|\boldsymbol{\theta}_0 - \boldsymbol{\theta}_1\| + L\epsilon_i\|\boldsymbol{\theta} - \boldsymbol{\theta}'\|. \tag{20}$$

By A4, the update recursion for the average iterate of DSGD-GD can be expressed as

$$\overline{\boldsymbol{\theta}}^{t+1} = (1/n)\sum_{i=1}^n \boldsymbol{\theta}_i^{t+1} = \overline{\boldsymbol{\theta}}^t - \gamma_{t+1}\sum_{i=1}^n \nabla\ell(\boldsymbol{\theta}_i^t; Z_i^{t+1})/n. \tag{21}$$

Using the above recursion, we show the following lemma for the one-step progress of DSGD-GD:

**Lemma 3. (Descent Lemma)** *Fix any $\delta > 0$ and let $\epsilon_{\mathsf{avg}} \leq \frac{\mu}{(1+\delta)L}$. Under A1, A2, A3, A5 and let the step sizes satisfy $\sup_{t\geq 0}\gamma_{t+1} \leq \frac{\widetilde{\mu}}{c_2}$, the following bound holds*

$$\mathbb{E}_t\left\|\widetilde{\boldsymbol{\theta}}^{t+1}\right\|^2 \leq (1 - \widetilde{\mu}\gamma_{t+1})\left\|\widetilde{\boldsymbol{\theta}}^t\right\|^2 + \left[c_1\gamma_{t+1} + c_2\gamma_{t+1}^2\right]\frac{1}{n}\left\|\boldsymbol{\Theta}_o^t\right\|_F^2 + \frac{2\sigma^2}{n}\gamma_{t+1}^2, \tag{22}$$

*for any $t \geq 0$, where $\mathbb{E}_t[\cdot]$ is the expectation operator conditioned on the iterates up to the $t$th iteration, and we recall the definitions of $c_1, c_2, \widetilde{\mu}$ from (14).*

The proof is in §B. We highlight that proving the upper bound (22) requires the smoothness property Lemma 2 for handling the difference $\sum_{i=1}^n \nabla f_i(\boldsymbol{\theta}_i^t, \boldsymbol{\theta}_i^t) - \nabla f_i(\boldsymbol{\theta}^{PS}, \boldsymbol{\theta}^{PS})$ as proportional to the error against the Multi-PS solution $\widetilde{\boldsymbol{\theta}}^t$ and the consensus error $\|\boldsymbol{\Theta}_o^t\|_F^2$. Lemma 3 prompts us to study the consensus error $\|\boldsymbol{\Theta}_o^t\|_F^2$ and a key observation is:

**Lemma 4. (Consensus Error Bound)** *Under A2–A6 and let the step sizes satisfy* $\sup_{t\geq 0} \gamma_{t+1} \leq \rho/\sqrt{2c_3}$, *then it holds*

$$\mathbb{E}_t[\tfrac{1}{n}\left\|\mathbf{\Theta}_o^{t+1}\right\|_F^2] \leq \left(1 - \frac{\rho}{2}\right)\tfrac{1}{n}\left\|\mathbf{\Theta}_o^t\right\|_F^2 + 12[\sigma^2 + \varsigma^2]\frac{\gamma_{t+1}^2}{\rho}\left\|\widetilde{\boldsymbol{\theta}}^t\right\|^2 + 9(\sigma^2 + \varsigma^2)\frac{\gamma_{t+1}^2}{\rho}, \quad (23)$$

*for any* $t \geq 0$, *where we recall that* $c_3 := 12\sigma^2 + 18L^2(1 + \epsilon_{\mathsf{max}})^2$.

The proof is in §C. In (52) of the appendix, we provide an alternative consensus error bound without using A6. Note that despite the decision dependent distributions due to the performative nature of Multi-PfD, the above bound shows a similar trend as in [Bars et al., 2022, Pu et al., 2021, Kong et al., 2021, Koloskova et al., 2019].

However, unlike [Bars et al., 2022, Lemma 2], the r.h.s. of (23) contains a $\mathcal{O}([\sigma^2 + \varsigma^2]\gamma_{t+1}^2\|\widetilde{\boldsymbol{\theta}}^t\|^2)$ term which arises from A5, A6 with the growth condition. This introduces new challenges to analysis as it will be insufficient to conclude from (23) *alone* that DSGD-GD converges to a consensual solution.

Our plan is to consider Lemmas 3 and 4 simultaneously in order to control $\mathbb{E}\|\widetilde{\boldsymbol{\theta}}^t\|^2$, $\mathbb{E}\|\mathbf{\Theta}_o^t\|_F^2$. Define the following sequence of non-negative numbers: for any $t \geq 0$,

$$\mathcal{L}_{t+1} := \mathbb{E}\big[\|\widetilde{\boldsymbol{\theta}}^{t+1}\|^2 + \gamma_{t+1}\frac{8c_1}{\rho n}\left\|\mathbf{\Theta}_o^{t+1}\right\|_F^2\big]. \quad (24)$$

We obtain the following lemma:

**Lemma 5 (Convergence of $\mathcal{L}_t$).** *Under A1–A6. Suppose that the step sizes satisfy* $\sup_{t\geq 0}\gamma_{t+1} \leq \min\left\{\frac{4}{\widetilde{\mu}}, \sqrt{\frac{\rho^2\widetilde{\mu}}{192c_1(\sigma^2+\varsigma^2)}}, \frac{\rho c_1}{4\widetilde{\mu}c_1 + \rho c_2}\right\}$. *For any* $t \geq 0$, *it holds*

$$\mathcal{L}_{t+1} \leq (1 - \widetilde{\mu}\gamma_{t+1}/2)\,\mathcal{L}_t + \rho^{-2}72c_1(\sigma^2 + \varsigma^2)\gamma_{t+1}^3 + n^{-1}2\sigma^2\gamma_{t+1}^2. \quad (25)$$

*Further, if the step sizes satisfy* $\frac{\gamma_{t-1}}{\gamma_t} \leq \min\{\sqrt{1 + (\widetilde{\mu}/4)\gamma_t^2}, \sqrt[3]{1 + (\widetilde{\mu}/4)\gamma_t^3}\}$ *for any* $t \geq 1$, *then*

$$\mathbb{E}\left[\left\|\widetilde{\boldsymbol{\theta}}^{t+1}\right\|^2 + \gamma_{t+1}\frac{8c_1}{\rho n}\left\|\mathbf{\Theta}_o^{t+1}\right\|_F^2\right] \leq \prod_{i=1}^{t+1}\left(1 - \frac{\widetilde{\mu}\gamma_i}{2}\right)\mathsf{D} + \frac{288c_1(\sigma^2 + \varsigma^2)}{\rho^2\widetilde{\mu}}\gamma_{t+1}^2 + \frac{8\sigma^2}{\widetilde{\mu}n}\gamma_{t+1}, \quad (26)$$

*where we recall that* $\mathsf{D} := \|\widetilde{\boldsymbol{\theta}}^0\|^2 + \gamma_1\frac{8c_1}{\rho n}\left\|\mathbf{\Theta}_o^0\right\|_F^2$.

The proof is in §D, where (25) is based on a careful combination of Lemmas 3 and 4, and (26) is computed from the non-asymptotic analysis for the recursion (25).

**Proof of Theorem 1.** Lemma 5 immediately leads to (16) of the theorem as the l.h.s. of (26) is lower bounded by $\mathbb{E}\|\widetilde{\boldsymbol{\theta}}^{t+1}\|^2$. To obtain (17), we observe from the simplifying the r.h.s. of (26) that

$$\sup_{t\geq 1}\mathbb{E}\left\|\widetilde{\boldsymbol{\theta}}^t\right\|^2 \leq \mathsf{D} + \frac{288c_1(\sigma^2 + \varsigma^2)}{\rho^2\widetilde{\mu}}\gamma_1^2 + \frac{8\sigma^2}{\widetilde{\mu}n}\gamma_1 \leq \mathsf{D} + \frac{3}{2} + \frac{8\sigma^2}{c_2 n} =: \overline{\Delta}. \quad (27)$$

Note that $\|\widetilde{\boldsymbol{\theta}}^0\|^2 \leq \overline{\Delta}$ as well. Substituting the above into Lemma 4 yields

$$\tfrac{1}{n}\mathbb{E}\left\|\mathbf{\Theta}_o^{t+1}\right\|_F^2 \leq (1 - \rho/2)\tfrac{1}{n}\mathbb{E}\left\|\mathbf{\Theta}_o^t\right\|_F^2 + \rho^{-1}(9 + 12\overline{\Delta})(\sigma^2 + \varsigma^2)\gamma_{t+1}^2$$

$$\leq \left(1 - \frac{\rho}{2}\right)^{t+1}\tfrac{1}{n}\left\|\mathbf{\Theta}_o^0\right\|_F^2 + \frac{(9 + 12\overline{\Delta})(\sigma^2 + \varsigma^2)}{\rho}\sum_{s=1}^{t+1}\left(1 - \frac{\rho}{2}\right)^{t+1-s}\gamma_s^2 \quad (28)$$

Applying Lemma 7 in the appendix together with the step size condition $\gamma_t/\gamma_{t+1} \leq 1 + \rho/(4 - 2\rho)$, $t \geq 1$, leads to (17) of the theorem. This concludes our proof. $\qquad\square$

## 4 Numerical Experiments

We consider two examples of performative prediction problems to verify our theories. All experiments are conducted with Python on a server using 80 threads of an Intel Xeon 6318 CPU.

**Multi-agent Gaussian Mean Estimation.** We aim to illustrate Proposition 1, Theorem 1 via a scalar Gaussian mean estimation problem on synthetic data. We consider $n = 25$ agents connected on a

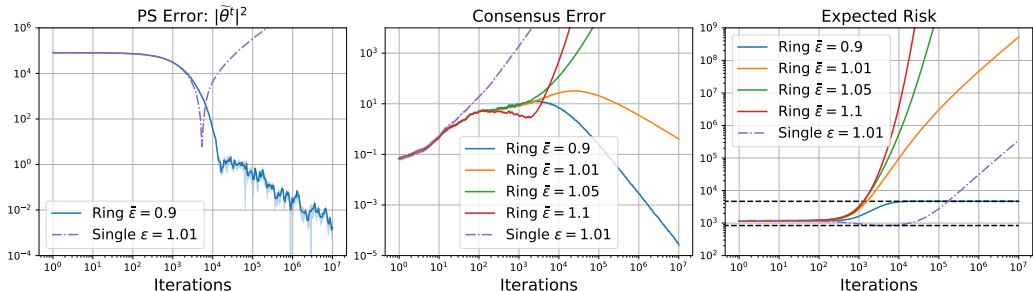

Figure 2: **Multi-agent Gaussian Mean Estimation.** (Left) Gap to Multi-PS solution. Note that $\boldsymbol{\theta}^{PS}$ does not exist if $\epsilon_{\text{avg}} \geq 1$ and the plots are thus skipped. (Middle) Consensus error $\|\boldsymbol{\Theta}_o^t\|_F^2$. (Right) Performative risk (7). Results are averaged over 10 runs and shaded area is 90% confidence interval.

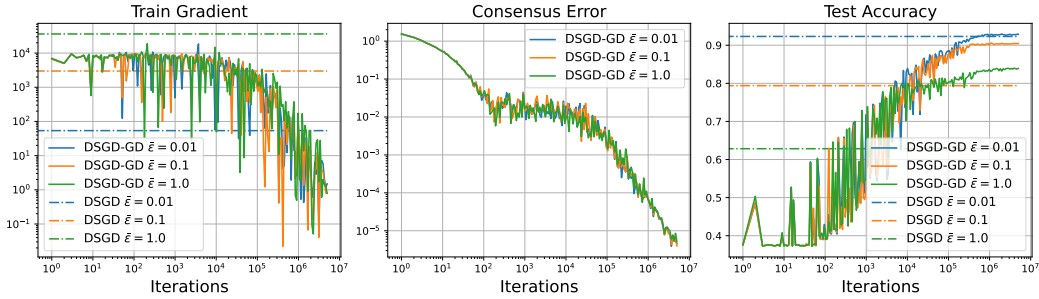

Figure 3: **Spam Email Classification.** (Left) Gradient on training dataset $\|\nabla f(\overline{\boldsymbol{\theta}}^t; \overline{\boldsymbol{\theta}}^t)\|^2$. Note that $\nabla f(\boldsymbol{\theta}^{PS}; \boldsymbol{\theta}^{PS}) = \mathbf{0}$ and thus the gradient norm measures the gap to $\boldsymbol{\theta}^{PS}$. (Middle) Consensus Error. (Right) Test accuracy with shifted distributions. We also compare the non-performative optimal solution (dashed lines, `DSGD` in the legend) on the shifted dataset.

ring graph, and the Multi-PfD problem (1) is specified with $\ell(\boldsymbol{\theta}_i; Z_i) = (\boldsymbol{\theta}_i - Z_i)^2/2$. The local distributions are given by $\mathcal{D}_i(\boldsymbol{\theta}_i) \equiv \mathcal{N}(\bar{z}_i + \epsilon_i \boldsymbol{\theta}_i, \sigma^2)$, where $\bar{z}_i$ is the mean value to be estimated. For this problem, we have $\mu = 1$, $L = 1$, as such if $0 < \bar{\epsilon} = \epsilon_{\text{avg}} < 1$, the Multi-PS solution can be computed in closed form as $\boldsymbol{\theta}^{PS} = \sum_{i=1}^n \bar{z}_i / [n(1 - \epsilon_{\text{avg}})]$; while $\boldsymbol{\theta}^{PS}$ does not exist if $\epsilon_{\text{avg}} \geq 1$.

In our experiments, we set $\bar{z}_i = 10$, $\sigma^2 = 50$ and step size for `DSGD-GD` as $\gamma_t = a_0/(a_1 + t)$ with $a_0 = 50$, $a_1 = 10^4$. In Fig. 2, we compare the gap $\|\overline{\boldsymbol{\theta}}^t - \boldsymbol{\theta}^{PS}\|^2$, consensus error $\|\boldsymbol{\Theta}_o^t\|^2$, expected performative risk $f(\overline{\boldsymbol{\theta}}^t; \overline{\boldsymbol{\theta}}^t)$ of (1), against the iteration number $t$. We examine the behavior of `DSGD-GD` when the Multi-PfD problem has an averaged sensitivity parameter of $\epsilon_{\text{avg}} \in \{0.9, 1.01, 1.05, 1.1\}$ and under a heterogeneous, decision-dependent distribution environment where $\epsilon_i$ are distinct.

We first observe from Fig. 2 (left) and (middle) that when $\epsilon_{\text{avg}} = 0.9 < 1$, the gap $\|\overline{\boldsymbol{\theta}}^t - \boldsymbol{\theta}^{PS}\|^2$ decays at $\mathcal{O}(1/t)$ as $t \to \infty$, while the consensus error $\|\boldsymbol{\Theta}_o^t\|^2$ decays at $\mathcal{O}(1/t^2)$. This coincides exactly with Theorem 1. Furthermore, in Fig. 2 (dash-dotted), we simulate a setting where $\epsilon_{\text{avg}} = 0.9$ but one of the agents with $\epsilon_i = 1.01$ is always disconnected from the network and perform the greedy deployment scheme *individually*. We observe from the figure that its performative risk $f_i(\boldsymbol{\theta}_i^t; \boldsymbol{\theta}_i^t)$ diverges as $t \to \infty$. This indicates that consensus can help stabilize the system.

Lastly, from Fig. 2 (middle) and (right), we observe that whenever $\epsilon_{\text{avg}} > 1$, the consensus error and performative risk diverge. Again, this corroborates with our Proposition 1.

**Email Spam Classification.** We evaluate the performance of `DSGD-GD` by simulating the performative effects on a real dataset. This example is a multi-agent spam classification task based on `spambase`, a dataset [Hopkins, 1999] with $m = 4601$ samples, $d = 48$ features. We adopt Example 1 and simulate a scenario with 25 regional servers on a ring graph. Each server has access to training data from $m_i = 138$ samples from `spambase` modeling the different set of users; the rest of $m_{\text{train}} = 1150$ samples are taken as testing data. The servers aim to find a common *spam filter classifier* via (2) with $\beta = 10^{-4}$. To model the strategic behavior of users, their features $\boldsymbol{X}_i$ are adapted to $\boldsymbol{\theta}_i$ through

maximizing a linear utility function, resulting in the shifted distribution $\mathcal{D}_i(\boldsymbol{\theta}_i)$ specified in (3). The sensitivity parameters are set as $\epsilon_i \in \{0.4\epsilon_{\mathsf{avg}}, 0.45\epsilon_{\mathsf{avg}}, \ldots, 1.6\epsilon_{\mathsf{avg}}\}$ with $\bar{\epsilon} = \epsilon_{\mathsf{avg}} \in \{0.01, 0.1, 1\}$.

Our results are shown in Fig. 3 as we compare the gradient $\|\nabla f(\overline{\boldsymbol{\theta}}^t; \overline{\boldsymbol{\theta}}^t)\|^2$ evaluated on the training dataset, the consensus error $\|\boldsymbol{\Theta}_o^t\|_F^2$, and the accuracy on the testing dataset, against the iteration number $t$. From Fig. 3 (left) and (middle), we observe that the DSGD-GD scheme converges to the Multi-PS solution and reaches consensus under various settings of $\epsilon_{\mathsf{avg}}$, at the rates $\mathcal{O}(1/t), \mathcal{O}(1/t^2)$, respectively. In Fig. 3 (right), we evaluate the performance of the trained classifier $\boldsymbol{\theta}_i^t$ on the testing dataset with shifted distribution due to $\boldsymbol{\theta}_i^t$. We compare with a non-performatively trained solution obtained by solving $\boldsymbol{\theta}^\star = \arg\min_{\boldsymbol{\theta}} \sum_{i=1}^n f_i(\boldsymbol{\theta}; \mathbf{0})$ (DSGD in the legend), i.e., without any shift in distributions, but evaluate the performance on distribution shifted by $\boldsymbol{\theta}^\star$. As observed, the test accuracy decreases as sensitivity $\epsilon_{\mathsf{avg}}$ increases, and DSGD-GD achieves better accuracy than DSGD.

## 5 Conclusions & Limitations

In this paper, we studied the Multi-PfD problem, and analyzed its stability when a DSGD-GD scheme is applied. Our results indicate that when agents are *consensus seeking*, Multi-PfD admits a performative stable solution with laxer condition and DSGD-GD achieves linear speedup. Limitations to our current results include the requirement of synchronous updates among agents, strongly convex loss [cf. A1], etc., which shall be explored in future extensions.

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
