# A Proof of Proposition 1

Fix any $\boldsymbol{\theta}', \boldsymbol{\theta} \in \mathbb{R}^d$. The optimality condition to (11) implies that

$$\sum_{i=1}^n \nabla f_i(\mathcal{M}(\boldsymbol{\theta}); \boldsymbol{\theta}) = \mathbf{0}, \qquad \sum_{i=1}^n \nabla f_i(\mathcal{M}(\boldsymbol{\theta}'); \boldsymbol{\theta}') = \mathbf{0}. \tag{29}$$

Note that the gradients are taken w.r.t. the first argument in the function $f_i$. Observe the chain

$$0 = \langle \mathbf{0} \,|\, \mathcal{M}(\boldsymbol{\theta}) - \mathcal{M}(\boldsymbol{\theta}') \rangle = \left\langle \sum_{i=1}^n \left[ \nabla f_i(\mathcal{M}(\boldsymbol{\theta}); \boldsymbol{\theta}) - \nabla f_i(\mathcal{M}(\boldsymbol{\theta}'); \boldsymbol{\theta}') \right] \,|\, \mathcal{M}(\boldsymbol{\theta}) - \mathcal{M}(\boldsymbol{\theta}') \right\rangle.$$

Adding and subtracting $\sum_{i=1}^n \nabla f_i(\mathcal{M}(\boldsymbol{\theta}); \boldsymbol{\theta}')$ implies the equality:

$$\begin{aligned}
&\sum_{i=1}^n \langle \nabla f_i(\mathcal{M}(\boldsymbol{\theta}); \boldsymbol{\theta}') - \nabla f_i(\mathcal{M}(\boldsymbol{\theta}); \boldsymbol{\theta}) \,|\, \mathcal{M}(\boldsymbol{\theta}) - \mathcal{M}(\boldsymbol{\theta}') \rangle \\
&= \sum_{i=1}^n \langle (\nabla f_i(\mathcal{M}(\boldsymbol{\theta}); \boldsymbol{\theta}') - \nabla f_i(\mathcal{M}(\boldsymbol{\theta}'); \boldsymbol{\theta}')) \,|\, \mathcal{M}(\boldsymbol{\theta}) - \mathcal{M}(\boldsymbol{\theta}') \rangle.
\end{aligned} \tag{30}$$

Applying A1 to the right hand side of (30) lead to:

$$\sum_{i=1}^n \langle (\nabla f_i(\mathcal{M}(\boldsymbol{\theta}); \boldsymbol{\theta}') - \nabla f_i(\mathcal{M}(\boldsymbol{\theta}'); \boldsymbol{\theta}')) \,|\, \mathcal{M}(\boldsymbol{\theta}) - \mathcal{M}(\boldsymbol{\theta}') \rangle \geq n\mu \|\mathcal{M}(\boldsymbol{\theta}) - \mathcal{M}(\boldsymbol{\theta}')\|^2.$$

Meanwhile, applying Lemma 2 to the left hand side of (30) gives

$$\begin{aligned}
&\sum_{i=1}^n \langle \nabla f_i(\mathcal{M}(\boldsymbol{\theta}); \boldsymbol{\theta}') - \nabla f_i(\mathcal{M}(\boldsymbol{\theta}); \boldsymbol{\theta}) \,|\, \mathcal{M}(\boldsymbol{\theta}) - \mathcal{M}(\boldsymbol{\theta}') \rangle \\
&\leq \sum_{i=1}^n \epsilon_i L \|\boldsymbol{\theta}' - \boldsymbol{\theta}\| \|\mathcal{M}(\boldsymbol{\theta}) - \mathcal{M}(\boldsymbol{\theta}')\|.
\end{aligned}$$

Substituting back into (30) implies that

$$\|\mathcal{M}(\boldsymbol{\theta}) - \mathcal{M}(\boldsymbol{\theta}')\| \leq \frac{\sum_{i=1}^n \epsilon_i L}{n\mu} \|\boldsymbol{\theta} - \boldsymbol{\theta}'\| = \frac{\epsilon_{\mathsf{avg}} L}{\mu} \|\boldsymbol{\theta} - \boldsymbol{\theta}'\|. \tag{31}$$

Therefore, the map $\mathcal{M} : \mathbb{R}^d \to \mathbb{R}^d$ is a contraction if $\epsilon_{\mathsf{avg}} < \mu/L$. Subsequently, by the Banach fixed point theorem [Granas and Dugundji, 2003], the map $\mathcal{M}(\boldsymbol{\theta})$ admits a unique fixed point which is denoted as $\boldsymbol{\theta}^{PS}$.

To prove the converse, we consider the following instantiation of (11) with

$$\ell(\theta; Z) = \frac{1}{2}(\theta - Z)^2, \quad Z \sim \mathcal{D}_i(\theta) \iff Z \sim \mathcal{N}(\mu_i + \epsilon_i \theta, 1) \tag{32}$$

Note that the above satisfies A1 with $\mu = 1$, A2 with $L = 1$, A3 with $\epsilon_i$ for $i = 1, \ldots, n$. We consider a case where it holds $\epsilon_{\mathsf{avg}} \geq \mu/L = 1$. We also let $\mu_{\mathsf{avg}} := (1/n)\sum_{i=1}^n \mu_i \neq 0$.

We observe

$$\begin{aligned}
f_i(\theta'; \theta) &= \mathbb{E}_{Z \sim \mathcal{D}_i(\theta)} \left[ \frac{1}{2}(\theta' - Z)^2 \right] = \mathbb{E}_{\tilde{Z} \sim \mathcal{N}(0,1)} \left[ \frac{1}{2}(\theta' - \mu_i - \epsilon_i \theta - \tilde{Z})^2 \right] \\
&= \frac{1}{2}(\theta' - \mu_i - \epsilon_i \theta)^2 + \frac{1}{2}.
\end{aligned} \tag{33}$$

For any $\theta \in \mathbb{R}$, it can be shown that

$$\mathcal{M}(\theta) = \underset{\theta' \in \mathbb{R}}{\arg\min} \; \frac{1}{2n} \sum_{i=1}^n (\theta' - \mu_i - \epsilon_i \theta)^2 = \epsilon_{\mathsf{avg}} \theta + \mu_{\mathsf{avg}} \tag{34}$$

Thus, applying the map for $T$ times leads to

$$\mathcal{M}^T(\theta) = \epsilon_{\mathsf{avg}}^T \theta + \left( 1 + \epsilon_{\mathsf{avg}} + \cdots + \epsilon_{\mathsf{avg}}^{T-1} \right) \mu_{\mathsf{avg}} \tag{35}$$

Since $\epsilon_{\mathsf{avg}} > 1$ and $\mu_{\mathsf{avg}} \neq 0$, we have $\lim_{T \to \infty} |\mathcal{M}^T(\theta)| = \infty$ and the map is not a contraction.

# B Proof of Lemma 3

Recall that $\widetilde{\boldsymbol{\theta}}^t := \overline{\boldsymbol{\theta}}^t - \boldsymbol{\theta}^{PS}$ is the error of averaged decision at the $t$th iteration. Using (21), we have

$$\left\|\widetilde{\boldsymbol{\theta}}^{t+1}\right\|^2 = \left\|\widetilde{\boldsymbol{\theta}}^t\right\|^2 - \frac{2\gamma_{t+1}}{n}\left\langle \widetilde{\boldsymbol{\theta}}^t \mid \sum_{i=1}^{n} \nabla\ell(\boldsymbol{\theta}_i^t; Z_i^{t+1})\right\rangle + \frac{\gamma_{t+1}^2}{n^2}\left\|\sum_{i=1}^{n} \nabla\ell(\boldsymbol{\theta}_i^t; Z_i^{t+1})\right\|^2. \quad (36)$$

We consider taking the conditional expectation $\mathbb{E}_t[\cdot]$ on the both sides. Using the fixed point condition $\sum_{i=1}^{n} \nabla f_i(\boldsymbol{\theta}^{PS}; \boldsymbol{\theta}^{PS}) = \mathbf{0}$, we observe the following equivalent expression for the last term

$$\left\|\sum_{i=1}^{n} \nabla\ell(\boldsymbol{\theta}_i^t; Z_i^{t+1})\right\|^2 = \left\|\sum_{i=1}^{n} \left[\nabla\ell(\boldsymbol{\theta}_i^t; Z_i^{t+1}) - \nabla f_i(\boldsymbol{\theta}_i^t; \boldsymbol{\theta}_i^t) + \nabla f_i(\boldsymbol{\theta}_i^t; \boldsymbol{\theta}_i^t) - \nabla f_i(\boldsymbol{\theta}^{PS}; \boldsymbol{\theta}^{PS})\right]\right\|^2$$

Observe that $Z_i^{t+1}$, $i = 1, \ldots, n$ are independent r.v.s, taking the conditional expectation $\mathbb{E}_t[\cdot]$ yields the upper bound to the above term

$$\mathbb{E}_t\left\|\sum_{i=1}^{n} \nabla\ell(\boldsymbol{\theta}_i^t; Z_i^{t+1})\right\|^2$$

$$\leq 2\sum_{i=1}^{n} \mathbb{E}_t\left\|\nabla\ell(\boldsymbol{\theta}_i^t; Z_i^{t+1}) - \nabla f_i(\boldsymbol{\theta}_i^t; \boldsymbol{\theta}_i^t)\right\|^2 + 2n\sum_{i=1}^{n} \left\|\nabla f_i(\boldsymbol{\theta}_i^t; \boldsymbol{\theta}_i^t) - \nabla f_i(\boldsymbol{\theta}^{PS}; \boldsymbol{\theta}^{PS})\right\|^2 \quad (37)$$

$$\leq 2\sum_{i=1}^{n} \sigma^2(1 + \left\|\boldsymbol{\theta}_i^t - \boldsymbol{\theta}^{PS}\right\|^2) + 2n\sum_{i=1}^{n} L^2(1 + \epsilon_i)^2\left\|\boldsymbol{\theta}_i^t - \boldsymbol{\theta}^{PS}\right\|^2$$

$$\leq 2\sigma^2 n + 4n[\sigma^2 + nL^2(1 + \epsilon_{\mathsf{max}})^2]\left\|\widetilde{\boldsymbol{\theta}}^t\right\|^2 + 4[\sigma^2 + nL^2(1 + \epsilon_{\mathsf{max}})^2]\left\|\boldsymbol{\Theta}_o^t\right\|_F^2$$

where the first inequality is due to A5 and Lemma 2. We conclude that

$$\frac{1}{n^2}\mathbb{E}_t\left\|\sum_{i=1}^{n} \nabla\ell(\boldsymbol{\theta}_i^t; Z_i^{t+1})\right\|^2 \leq \frac{2\sigma^2}{n} + c_2\left\|\widetilde{\boldsymbol{\theta}}^t\right\|^2 + c_2\frac{1}{n}\left\|\boldsymbol{\Theta}_o^t\right\|_F^2 \quad (38)$$

where we recall the definition that $c_2 = 4\left(\frac{\sigma^2}{n} + L^2(1 + \epsilon_{\mathsf{max}})^2\right)$.

Next, we focus on the inner product term in (36), we have

$$\left\langle \widetilde{\boldsymbol{\theta}}^t \mid \sum_{i=1}^{n} \nabla f_i(\boldsymbol{\theta}_i^t, \boldsymbol{\theta}_i^t)\right\rangle = \sum_{i=1}^{n} \left\langle \widetilde{\boldsymbol{\theta}}^t \mid \nabla f_i(\boldsymbol{\theta}_i^t; \boldsymbol{\theta}_i^t) - \nabla f_i(\overline{\boldsymbol{\theta}}^t; \boldsymbol{\theta}^{PS})\right\rangle$$
$$+ \sum_{i=1}^{n} \left\langle \widetilde{\boldsymbol{\theta}}^t \mid \nabla f_i(\overline{\boldsymbol{\theta}}^t; \boldsymbol{\theta}^{PS}) - \nabla f_i(\boldsymbol{\theta}^{PS}; \boldsymbol{\theta}^{PS})\right\rangle \quad (39)$$

Applying the Cauchy-Schwarz inequality and A2, A3, we obtain

$$\sum_{i=1}^{n} \left\langle \widetilde{\boldsymbol{\theta}}^t \mid \nabla f_i(\boldsymbol{\theta}_i^t; \boldsymbol{\theta}_i^t) - \nabla f_i(\overline{\boldsymbol{\theta}}^t; \boldsymbol{\theta}^{PS})\right\rangle \geq -\left\|\widetilde{\boldsymbol{\theta}}^t\right\|\sum_{i=1}^{n}\left(L\left\|\boldsymbol{\theta}_i^t - \overline{\boldsymbol{\theta}}^t\right\| + L\varepsilon_i\left\|\boldsymbol{\theta}_i^t - \boldsymbol{\theta}^{PS}\right\|\right)$$

$$\geq -\left\|\widetilde{\boldsymbol{\theta}}^t\right\|\sum_{i=1}^{n}\left(L(1 + \epsilon_i)\left\|\boldsymbol{\theta}_i^t - \overline{\boldsymbol{\theta}}^t\right\| + L\epsilon_i\left\|\widetilde{\boldsymbol{\theta}}^t\right\|\right). \quad (40)$$

Meanwhile, using the strong convexity property of $\ell(\cdot; \cdot)$ [cf. A1], we have

$$\sum_{i=1}^{n} \left\langle \widetilde{\boldsymbol{\theta}}^t \mid \nabla f_i(\overline{\boldsymbol{\theta}}^t; \boldsymbol{\theta}^{PS}) - \nabla f_i(\boldsymbol{\theta}^{PS}; \boldsymbol{\theta}^{PS})\right\rangle \geq n\mu\left\|\widetilde{\boldsymbol{\theta}}^t\right\|^2. \quad (41)$$

Summing up the two lower bounds and rearranging terms give

$$\frac{1}{n}\mathbb{E}_t\left\langle \widetilde{\boldsymbol{\theta}}^t \mid \sum_{i=1}^{n} \nabla f_i(\boldsymbol{\theta}_i^t, \boldsymbol{\theta}_i^t)\right\rangle \geq (\mu - L\epsilon_{\mathsf{avg}})\left\|\widetilde{\boldsymbol{\theta}}^t\right\|^2 - \frac{L}{n}(1 + \epsilon_{\mathsf{max}})\sum_{i=1}^{n}\left\|\widetilde{\boldsymbol{\theta}}^t\right\|\left\|\boldsymbol{\theta}_i^t - \overline{\boldsymbol{\theta}}^t\right\|. \quad (42)$$

For any $\alpha > 0$, using the Young's inequality shows that the above can be further lower bounded by

$$
\begin{aligned}
&\left[\mu - L\epsilon_{\text{avg}} - \frac{\alpha}{2n}L(1+\epsilon_{\max})\right]\left\|\widetilde{\boldsymbol{\theta}}^t\right\|^2 - \frac{L(1+\epsilon_{\max})}{2n\alpha}\sum_{i=1}^n\left\|\boldsymbol{\theta}_i^t - \overline{\boldsymbol{\theta}}^t\right\|^2 \\
&\geq \left[\mu - L\epsilon_{avg} - \frac{\alpha}{2n}L(1+\epsilon_{\max})\right]\left\|\widetilde{\boldsymbol{\theta}}^t\right\|^2 - \frac{L(1+\epsilon_{\max})}{2n\alpha}\left\|\boldsymbol{\Theta}_o^t\right\|_F^2 \\
&\geq \left[\mu - (1+\delta)L\epsilon_{\text{avg}}\right]\left\|\widetilde{\boldsymbol{\theta}}^t\right\|^2 - \frac{L(1+\epsilon_{max})^2}{4n^2\delta\epsilon_{\text{avg}}}\left\|\boldsymbol{\Theta}_o^t\right\|_F^2,
\end{aligned}
\tag{43}
$$

where we have set $\alpha = \frac{2n\delta\epsilon_{\text{avg}}}{1+\epsilon_{\max}}$ to yield the last inequality.

Substituting (38), (43) back to the inequality (36) gives us the desired result. In particular,

$$
\begin{aligned}
\mathbb{E}_t\left\|\widetilde{\boldsymbol{\theta}}^{t+1}\right\|^2 &\leq \left\|\widetilde{\boldsymbol{\theta}}^t\right\|^2 - 2\gamma_{t+1}\left[[\mu - (1+\delta)L\epsilon_{\text{avg}}]\left\|\widetilde{\boldsymbol{\theta}}^t\right\|^2 - \frac{L(1+\epsilon_{max})^2}{4n^2\delta\epsilon_{\text{avg}}}\left\|\boldsymbol{\Theta}_o^t\right\|_F^2\right] \\
&\quad + \gamma_{t+1}^2\left[\frac{2\sigma^2}{n} + c_2\left\|\widetilde{\boldsymbol{\theta}}^t\right\|^2 + c_2\frac{1}{n}\left\|\boldsymbol{\Theta}_o^t\right\|_F^2\right] \\
&= \left(1 - 2\widetilde{\mu}\gamma_{t+1} + c_2\gamma_{t+1}^2\right)\left\|\widetilde{\boldsymbol{\theta}}^t\right\|^2 + \left[c_1\frac{\gamma_{t+1}}{n} + c_2\frac{\gamma_{t+1}^2}{n}\right]\left\|\boldsymbol{\Theta}_o^t\right\|_F^2 + \frac{2\sigma^2}{n}\gamma_{t+1}^2 \\
&\leq \left(1 - \widetilde{\mu}\gamma_{t+1}\right)\left\|\widetilde{\boldsymbol{\theta}}^t\right\|^2 + \left[c_1\frac{\gamma_{t+1}}{n} + c_2\frac{\gamma_{t+1}^2}{n}\right]\left\|\boldsymbol{\Theta}_o^t\right\|_F^2 + \frac{2\sigma^2}{n}\gamma_{t+1}^2
\end{aligned}
\tag{44}
$$

where we recall the constants $c_1 := \frac{L(1+\epsilon_{\max})^2}{2n\delta\epsilon_{\text{avg}}}$, $c_2 := 4\left(\frac{\sigma^2}{n} + L^2(1+\epsilon_{\max})^2\right)$ and $\widetilde{\mu} := \mu - (1+\delta)\epsilon_{\text{avg}}L$ and the last inequality is obtained by observing the condition $\gamma_{t+1} \leq \widetilde{\mu}/c_2$.

## C   Proof of Lemma 4

To simplify notations, we denote

$$
\begin{aligned}
\widetilde{\nabla}F^t &:= \left(\nabla\ell(\boldsymbol{\theta}_1^t; Z_1^{t+1}), \cdots, \nabla\ell(\boldsymbol{\theta}_n^t; Z_n^{t+1})\right)^\top \in \mathbb{R}^{n\times d}, \\
\boldsymbol{\Theta}^t &:= \left(\boldsymbol{\theta}_1^t, \cdots, \boldsymbol{\theta}_n^t\right)^\top \in \mathbb{R}^{n\times d}, \quad \overline{\boldsymbol{\Theta}}^t := (1/n)\mathbf{1}\mathbf{1}^\top\boldsymbol{\Theta}^t \in \mathbb{R}^n.
\end{aligned}
\tag{45}
$$

Notice that $\boldsymbol{\Theta}_o^t = \boldsymbol{\Theta}^t - \overline{\boldsymbol{\Theta}}^t = (\boldsymbol{I} - (1/n)\mathbf{1}\mathbf{1}^\top)\boldsymbol{\Theta}^t$. We first observe the following relation:

$$
\begin{aligned}
\boldsymbol{\Theta}_o^{t+1} &= \boldsymbol{\Theta}^{t+1} - \overline{\boldsymbol{\Theta}}^{t+1} = \left(\boldsymbol{I} - \frac{1}{n}\mathbf{1}\mathbf{1}^\top\right)\boldsymbol{\Theta}^{t+1} = \left(\boldsymbol{I} - \frac{1}{n}\mathbf{1}\mathbf{1}^\top\right)\left(\boldsymbol{W}\boldsymbol{\Theta}^t - \gamma_{t+1}\widetilde{\nabla}F^t\right) \\
&= \left(\boldsymbol{W} - \frac{1}{n}\mathbf{1}\mathbf{1}^\top\right)\boldsymbol{\Theta}_o^t - \gamma_{t+1}\left(\boldsymbol{I} - \frac{1}{n}\mathbf{1}\mathbf{1}^\top\right)\widetilde{\nabla}F^t,
\end{aligned}
$$

where the last equality is due to $(\boldsymbol{I} - (1/n)\mathbf{1}\mathbf{1}^\top)\boldsymbol{W} = (\boldsymbol{W} - (1/n)\mathbf{1}\mathbf{1}^\top)(\boldsymbol{I} - (1/n)\mathbf{1}\mathbf{1}^\top)$ as $\boldsymbol{W}$ is a doubly stochastic matrix.

Computing the squared norm of the consensus error leads to: for any $\alpha > 0$,

$$
\begin{aligned}
\mathbb{E}_t\left\|\boldsymbol{\Theta}_o^{t+1}\right\|_F^2 &\leq (1+\alpha)(1-\rho)^2\left\|\boldsymbol{\Theta}_o^t\right\|_F^2 + (1+\frac{1}{\alpha})\gamma_{t+1}^2\mathbb{E}_t\left\|\left(\boldsymbol{I} - \frac{1}{n}\mathbf{1}\mathbf{1}^\top\right)\widetilde{\nabla}F^t\right\|_F^2 \\
&\leq (1-\rho)\left\|\boldsymbol{\Theta}_o^t\right\|_F^2 + \frac{\gamma_{t+1}^2}{\rho}\mathbb{E}_t\left\|\left(\boldsymbol{I} - \frac{1}{n}\mathbf{1}\mathbf{1}^\top\right)\widetilde{\nabla}F^t\right\|_F^2,
\end{aligned}
\tag{46}
$$

where we have applied A4 in the first inequality and set $\alpha = \frac{\rho}{1-\rho}$ in the second inequality. The last term in the above inequality can be bounded as

$$
\mathbb{E}_t \left\| \left( \boldsymbol{I} - \frac{1}{n} \mathbf{1}\mathbf{1}^\top \right) \widetilde{\nabla} F^t \right\|_F^2 = \mathbb{E}_t \left[ \sum_{i=1}^n \left\| \nabla \ell(\boldsymbol{\theta}_i^t; Z_i^{t+1}) - \frac{1}{n} \sum_{j=1}^n \nabla \ell(\boldsymbol{\theta}_j^t; Z_j^{t+1}) \right\|^2 \right]
$$

$$
\leq 3 \sum_{i=1}^n \mathbb{E}_t \left\| \nabla \ell(\boldsymbol{\theta}_i^t; Z_i^{t+1}) - \nabla f_i(\boldsymbol{\theta}_i^t, \boldsymbol{\theta}_i^t) \right\|^2 + \frac{3}{n} \sum_{j=1}^n \mathbb{E}_t \left\| \nabla \ell(\boldsymbol{\theta}_j^t; Z_j^{t+1}) - \nabla f_j(\boldsymbol{\theta}_j^t, \boldsymbol{\theta}_j^t) \right\|^2
$$

$$
+ 3 \sum_{i=1}^n \left\| \nabla f_i(\boldsymbol{\theta}_i^t, \boldsymbol{\theta}_i^t) - \frac{1}{n} \sum_{j=1}^n \nabla f_j(\boldsymbol{\theta}_j^t, \boldsymbol{\theta}_j^t) \right\|^2 \tag{47}
$$

$$
\leq 6\sigma^2 \left( n + \sum_{i=1}^n \left\| \boldsymbol{\theta}_i^t - \boldsymbol{\theta}^{PS} \right\|^2 \right) + 3 \sum_{i=1}^n \left\| \nabla f_i(\boldsymbol{\theta}_i^t, \boldsymbol{\theta}_i^t) - \frac{1}{n} \sum_{j=1}^n \nabla f_j(\boldsymbol{\theta}_j^t, \boldsymbol{\theta}_j^t) \right\|^2
$$

$$
\leq 6\sigma^2 \left( n + 2n \left\| \widetilde{\boldsymbol{\theta}}^t \right\|^2 + 2 \left\| \boldsymbol{\Theta}_o^t \right\|_F^2 \right) + 3 \sum_{i=1}^n \left\| \nabla f_i(\boldsymbol{\theta}_i^t, \boldsymbol{\theta}_i^t) - \frac{1}{n} \sum_{j=1}^n \nabla f_j(\boldsymbol{\theta}_j^t, \boldsymbol{\theta}_j^t) \right\|^2
$$

where the second last inequality is due to A5. For each $i = 1, \ldots, n$, we observe

$$
\left\| \nabla f_i(\boldsymbol{\theta}_i^t, \boldsymbol{\theta}_i^t) - \nabla f_i(\overline{\boldsymbol{\theta}}^t, \overline{\boldsymbol{\theta}}^t) + \nabla f_i(\overline{\boldsymbol{\theta}}^t, \overline{\boldsymbol{\theta}}^t) - \frac{1}{n} \sum_{j=1}^n \nabla f_j(\overline{\boldsymbol{\theta}}^t, \overline{\boldsymbol{\theta}}^t) - \frac{1}{n} \sum_{j=1}^n [\nabla f_j(\boldsymbol{\theta}_j^t, \boldsymbol{\theta}_j^t) - \nabla f_j(\overline{\boldsymbol{\theta}}^t, \overline{\boldsymbol{\theta}}^t)] \right\|^2
$$

$$
\leq 3 \left\| \nabla f_i(\boldsymbol{\theta}_i^t, \boldsymbol{\theta}_i^t) - \nabla f_i(\overline{\boldsymbol{\theta}}^t, \overline{\boldsymbol{\theta}}^t) \right\|^2 + 3 \left\| \nabla f_i(\overline{\boldsymbol{\theta}}^t, \overline{\boldsymbol{\theta}}^t) - \frac{1}{n} \sum_{j=1}^n \nabla f_j(\overline{\boldsymbol{\theta}}^t, \overline{\boldsymbol{\theta}}^t) \right\|^2 \tag{48}
$$

$$
+ \frac{3}{n} \sum_{j=1}^n \left\| \nabla f_j(\boldsymbol{\theta}_j^t, \boldsymbol{\theta}_j^t) - \nabla f_j(\overline{\boldsymbol{\theta}}^t, \overline{\boldsymbol{\theta}}^t) \right\|^2
$$

$$
\leq 3 \left\| \nabla f_i(\boldsymbol{\theta}_i^t, \boldsymbol{\theta}_i^t) - \nabla f_i(\overline{\boldsymbol{\theta}}^t, \overline{\boldsymbol{\theta}}^t) \right\|^2 + \frac{3}{n} \sum_{j=1}^n \left\| \nabla f_j(\boldsymbol{\theta}_j^t, \boldsymbol{\theta}_j^t) - \nabla f_j(\overline{\boldsymbol{\theta}}^t, \overline{\boldsymbol{\theta}}^t) \right\|^2 + 3\varsigma^2 \left( 1 + \left\| \widetilde{\boldsymbol{\theta}}^t \right\|^2 \right)
$$

where the last inequality is due to A6. Now, we observe

$$
\sum_{i=1}^n \left\| \nabla f_i(\boldsymbol{\theta}_i^t, \boldsymbol{\theta}_i^t) - \frac{1}{n} \sum_{j=1}^n \nabla f_j(\boldsymbol{\theta}_j^t, \boldsymbol{\theta}_j^t) \right\|^2 \leq 6 \sum_{i=1}^n \left\| \nabla f_i(\boldsymbol{\theta}_i^t, \boldsymbol{\theta}_i^t) - \nabla f_i(\overline{\boldsymbol{\theta}}^t, \overline{\boldsymbol{\theta}}^t) \right\|^2 + 3n\varsigma^2 \left( 1 + \left\| \widetilde{\boldsymbol{\theta}}^t \right\|^2 \right)
$$

$$
\leq 6L^2(1 + \epsilon_{\mathsf{max}})^2 \left\| \boldsymbol{\Theta}_o^t \right\|_F^2 + 3n\varsigma^2 \left( 1 + \left\| \widetilde{\boldsymbol{\theta}}^t \right\|^2 \right)
$$

where the second inequality is due to Lemma 2 and the definition of $\boldsymbol{\Theta}_o^t$.

Substituting the above bounds into (47) leads to

$$
\mathbb{E}_t \left\| \left( \boldsymbol{I} - \frac{1}{n} \mathbf{1}\mathbf{1}^\top \right) \widetilde{\nabla} F^t \right\|_F^2
$$

$$
\leq 6\sigma^2 \left( n + 2n \left\| \widetilde{\boldsymbol{\theta}}^t \right\|^2 + 2 \left\| \boldsymbol{\Theta}_o^t \right\|_F^2 \right) + 18L^2(1 + \epsilon_{\mathsf{max}})^2 \left\| \boldsymbol{\Theta}_o^t \right\|_F^2 + 9n\varsigma^2 \left( 1 + \left\| \widetilde{\boldsymbol{\theta}}^t \right\|^2 \right) \tag{49}
$$

$$
\leq 9n[\sigma^2 + \varsigma^2] + 12n[\sigma^2 + \varsigma^2] \left\| \widetilde{\boldsymbol{\theta}}^t \right\|^2 + [12\sigma^2 + 18L^2(1 + \epsilon_{\mathsf{max}})^2] \left\| \boldsymbol{\Theta}_o^t \right\|_F^2
$$

Let $c_3 := 12\sigma^2 + 18L^2(1 + \epsilon_{\max})^2$. Substituting the above inequality into (46) gives us

$$\mathbb{E}_t \left\|\boldsymbol{\Theta}_o^{t+1}\right\|_F^2 \le (1 - \rho)\left\|\boldsymbol{\Theta}_o^t\right\|_F^2 + \frac{\gamma_{t+1}^2}{\rho}\left(12n[\sigma^2 + \varsigma^2]\left\|\widetilde{\boldsymbol{\theta}}^t\right\|^2 + c_3\left\|\boldsymbol{\Theta}_o^t\right\|_F^2\right) + 9n(\sigma^2 + \varsigma^2)\frac{\gamma_{t+1}^2}{\rho}$$

$$\le (1 - \rho/2)\left\|\boldsymbol{\Theta}_o^t\right\|_F^2 + \frac{\gamma_{t+1}^2}{\rho}12n[\sigma^2 + \varsigma^2]\left\|\widetilde{\boldsymbol{\theta}}^t\right\|^2 + 9n(\sigma^2 + \varsigma^2)\frac{\gamma_{t+1}^2}{\rho},$$

where the last inequality is due to the step size condition $\gamma_{t+1}^2 \le \rho^2/2c_3$. The proof is concluded.

**Alternative Bound without A6** We consider bounding (48) without using A6. Instead, we only assume that $\max_{i=1,\ldots,n}\left\|\nabla f_i(\boldsymbol{\theta}^{PS}; \boldsymbol{\theta}^{PS})\right\|^2 \le \varsigma^2$. We observe

$$\left\|\nabla f_i(\overline{\boldsymbol{\theta}}^t, \overline{\boldsymbol{\theta}}^t) - \nabla f(\overline{\boldsymbol{\theta}}^t, \overline{\boldsymbol{\theta}}^t)\right\|^2 \le 2\left\|\nabla f_i(\boldsymbol{\theta}^{PS}; \boldsymbol{\theta}^{PS})\right\|^2$$

$$+ 2\left\|\nabla f_i(\overline{\boldsymbol{\theta}}^t, \overline{\boldsymbol{\theta}}^t) - \nabla f_i(\boldsymbol{\theta}^{PS}; \boldsymbol{\theta}^{PS}) + \nabla f(\boldsymbol{\theta}^{PS}; \boldsymbol{\theta}^{PS}) - \nabla f(\overline{\boldsymbol{\theta}}^t; \overline{\boldsymbol{\theta}}^t)\right\|^2 \quad (50)$$

$$\le 2\left\|\nabla f_i(\boldsymbol{\theta}^{PS}; \boldsymbol{\theta}^{PS})\right\|^2 + 8L^2(1 + \epsilon_{\max})^2\left\|\widetilde{\boldsymbol{\theta}}^t\right\|^2 \le 2\varsigma^2 + 8L^2(1 + \epsilon_{\max})^2\left\|\widetilde{\boldsymbol{\theta}}^t\right\|^2,$$

for all $i = 1, \ldots, n$. This leads to

$$\sum_{i=1}^n\left\|\nabla f_i(\boldsymbol{\theta}_i^t, \boldsymbol{\theta}_i^t) - \frac{1}{n}\sum_{j=1}^n\nabla f_j(\boldsymbol{\theta}_j^t, \boldsymbol{\theta}_j^t)\right\|^2$$

$$\le 6L^2(1 + \epsilon_{\max})^2\left\|\boldsymbol{\Theta}_o^t\right\|_F^2 + 2n\varsigma^2 + 8nL^2(1 + \epsilon_{\max})^2\left\|\widetilde{\boldsymbol{\theta}}^t\right\|^2.$$

Subsequently,

$$\mathbb{E}_t\left\|\left(\boldsymbol{I} - \frac{1}{n}\mathbf{1}\mathbf{1}^\top\right)\widetilde{\nabla}F^t\right\|_F^2$$

$$\le 6\sigma^2\left(n + 2n\left\|\widetilde{\boldsymbol{\theta}}^t\right\|^2 + 2\left\|\boldsymbol{\Theta}_o^t\right\|_F^2\right) + 6n\varsigma^2 + 6L^2(1 + \epsilon_{\max})^2\left(3\left\|\boldsymbol{\Theta}_o^t\right\|_F^2 + 4n\left\|\widetilde{\boldsymbol{\theta}}^t\right\|^2\right) \quad (51)$$

$$= 6n[\sigma^2 + \varsigma^2] + 12n\left[\sigma^2 + 2L^2(1 + \epsilon_{\max})^2\right]\left\|\widetilde{\boldsymbol{\theta}}^t\right\|^2 + \left[12\sigma^2 + 18L^2(1 + \epsilon_{\max})^2\right]\left\|\boldsymbol{\Theta}_o^t\right\|_F^2$$

Taking $c_3 := 12\sigma^2 + 18L^2(1 + \epsilon_{\max})^2$ as before and substituting the inequality into (46) yields

$$\mathbb{E}_t\left\|\boldsymbol{\Theta}_o^{t+1}\right\|_F^2$$

$$\le (1 - \rho)\left\|\boldsymbol{\Theta}_o^t\right\|_F^2 + \frac{\gamma_{t+1}^2}{\rho}\left(12n\left[\sigma^2 + 2L^2(1 + \epsilon_{\max})^2\right]\left\|\widetilde{\boldsymbol{\theta}}^t\right\|^2 + c_3\left\|\boldsymbol{\Theta}_o^t\right\|_F^2\right) + 6n(\sigma^2 + \varsigma^2)\frac{\gamma_{t+1}^2}{\rho}$$

$$\le (1 - \rho/2)\left\|\boldsymbol{\Theta}_o^t\right\|_F^2 + \frac{\gamma_{t+1}^2}{\rho}12n\left[\sigma^2 + 2L^2(1 + \epsilon_{\max})^2\right]\left\|\widetilde{\boldsymbol{\theta}}^t\right\|^2 + 6n(\sigma^2 + \varsigma^2)\frac{\gamma_{t+1}^2}{\rho},$$

where the last inequality is due to $\sup_{t \ge 1}\gamma_t \le \rho/\sqrt{2c_3}$. The above can be simplified into

$$\frac{1}{n}\mathbb{E}_t\left\|\boldsymbol{\Theta}_o^{t+1}\right\|_F^2 \le \left(1 - \frac{\rho}{2}\right)\frac{1}{n}\left\|\boldsymbol{\Theta}_o^t\right\|_F^2 + \frac{\gamma_{t+1}^2}{\rho}12\left[\sigma^2 + 2L^2(1 + \epsilon_{\max})^2\right]\left\|\widetilde{\boldsymbol{\theta}}^t\right\|^2 + 6(\sigma^2 + \varsigma^2)\frac{\gamma_{t+1}^2}{\rho}. \quad (52)$$

Compared to (23), we observe that the above bound entails a larger coefficient for $\|\widetilde{\boldsymbol{\theta}}^t\|^2$ which lead to a (slightly) worse convergence bound for the DSGD-GD scheme.

Lastly, we should mention that as in the original Lemma 4, (52) can also be combined with Lemma 3 to develop an alternate version of Lemma 5. Subsequently, we can achieve a similar result as Theorem 1 without assuming A6.

## D   Proof of Lemma 5

Combining Lemmas 3 and 4 leads to

$$\mathcal{L}_{t+1} \le (1 - \widetilde{\mu}\gamma_{t+1})\mathbb{E}\left\|\widetilde{\boldsymbol{\theta}}^t\right\|^2 + \left[c_1\gamma_{t+1} + c_2\gamma_{t+1}^2\right]\frac{1}{n}\mathbb{E}\left\|\boldsymbol{\Theta}_o^t\right\|_F^2 + \frac{2\sigma^2}{n}\gamma_{t+1}^2$$

$$+ \gamma_{t+1} \frac{8c_1}{\rho} \left( \left(1 - \frac{\rho}{2}\right) \frac{1}{n} \mathbb{E} \left\| \boldsymbol{\Theta}_o^t \right\|_F^2 + \frac{\gamma_{t+1}^2}{\rho} 12[\sigma^2 + \varsigma^2] \mathbb{E} \left\| \widetilde{\boldsymbol{\theta}}^t \right\|^2 + 9(\sigma^2 + \varsigma^2) \frac{\gamma_{t+1}^2}{\rho} \right)$$

$$= \left( 1 - \widetilde{\mu}\gamma_{t+1} + \frac{96c_1}{\rho^2} [\sigma^2 + \varsigma^2] \gamma_{t+1}^3 \right) \mathbb{E} \left\| \widetilde{\boldsymbol{\theta}}^t \right\|^2 + \frac{2\sigma^2}{n} \gamma_{t+1}^2 + \frac{72c_1}{\rho^2} (\sigma^2 + \varsigma^2) \gamma_{t+1}^3$$

$$+ \gamma_t \frac{8c_1}{\rho} \left( \frac{\gamma_{t+1}}{\gamma_t} \left(1 - \frac{\rho}{2}\right) + \frac{\rho}{8} + \frac{c_2 \rho}{8c_1} \gamma_{t+1} \right) \frac{1}{n} \mathbb{E} \left\| \boldsymbol{\Theta}_o^t \right\|_F^2$$

Note that by the step size conditions specified in the lemma, we have

$$1 - \widetilde{\mu}\gamma_{t+1} + \frac{96c_1}{\rho^2} [\sigma^2 + \varsigma^2] \gamma_{t+1}^3 \le 1 - \widetilde{\mu}\gamma_{t+1}/2$$

$$\frac{\gamma_{t+1}}{\gamma_t} \left(1 - \frac{\rho}{2}\right) + \frac{\rho}{8} + \frac{c_2 \rho}{8c_1} \gamma_{t+1} \le 1 - \widetilde{\mu}\gamma_{t+1}/2. \tag{53}$$

Thus, we obtain

$$\mathcal{L}_{t+1} \le (1 - \widetilde{\mu}\gamma_{t+1}/2)\mathcal{L}_t + \frac{2\sigma^2}{n} \gamma_{t+1}^2 + \frac{72c_1}{\rho^2} (\sigma^2 + \varsigma^2) \gamma_{t+1}^3. \tag{54}$$

This concludes the first part of the lemma, i.e., (25). For the second part, we further expand (54) to obtain

$$\mathcal{L}_{t+1} \le \prod_{i=1}^{t+1} \left(1 - \frac{\widetilde{\mu}\gamma_i}{2}\right) \mathsf{D} + \sum_{s=1}^{t+1} \prod_{i=s+1}^{t+1} (1 - \widetilde{\mu}\gamma_i/2) \left( \frac{2\sigma^2}{n} \gamma_s^2 + \frac{72c_1}{\rho^2} (\sigma^2 + \varsigma^2) \gamma_s^3 \right)$$

$$\le \prod_{i=1}^{t+1} \left(1 - \frac{\widetilde{\mu}\gamma_i}{2}\right) \mathsf{D} + \frac{288c_1(\sigma^2 + \varsigma^2)}{\rho^2 \widetilde{\mu}} \gamma_{t+1}^2 + \frac{8\sigma^2}{\widetilde{\mu}n} \gamma_{t+1}. \tag{55}$$

where we recall that $\mathsf{D} := \|\widetilde{\boldsymbol{\theta}}^0\|^2 + \frac{8\gamma_1 c_1}{\rho n} \left\| \boldsymbol{\Theta}_o^0 \right\|_F^2$ and the last inequality is due to Lemma 6 together with the specified step size conditions. The proof is thus concluded.

## E  Auxilliary Results

**Lemma 6.** *Consider a sequence of non-negative, non-increasing step sizes $\{\gamma_t\}_{t \ge 1}$. Let $a > 0$, $p \in \mathbb{Z}_+$ and $\gamma_1 < 2/a$. If $\gamma_t^p / \gamma_{t+1}^p \le 1 + (a/2)\gamma_{t+1}^p$ for any $t \ge 1$, then*

$$\sum_{j=1}^{t} \gamma_j^{p+1} \prod_{\ell=j+1}^{t} (1 - \gamma_\ell a) \le \frac{2}{a} \gamma_t^p, \quad \forall\, t \ge 1. \tag{56}$$

*Proof.* Observe that:

$$\sum_{j=1}^{t} \gamma_j^{p+1} \prod_{\ell=j+1}^{t} (1 - \gamma_\ell a) = \gamma_t^p \sum_{j=1}^{t} \gamma_j \prod_{\ell=j+1}^{t} \frac{\gamma_{\ell-1}^p}{\gamma_\ell^p} (1 - \gamma_\ell a)$$

$$\overset{(a)}{\le} \gamma_t^p \sum_{j=1}^{t} \gamma_j \prod_{\ell=j+1}^{t} \left(1 - \gamma_\ell \frac{a}{2}\right)$$

$$= \frac{2\gamma_t^p}{a} \sum_{j=1}^{t} \left( \prod_{\ell=j+1}^{t} (1 - \gamma_\ell a/2) - \prod_{\ell'=j}^{t} (1 - \gamma_{\ell'} a/2) \right)$$

$$= \frac{2\gamma_t^p}{a} \left( 1 - \prod_{\ell'=1}^{t} (1 - \gamma_{\ell'} a/2) \right) \le \frac{2\gamma_t^p}{a},$$

where (a) is due to the following observation

$$\frac{\gamma_{\ell-1}^p}{\gamma_\ell^p} (1 - \gamma_\ell a) \le \left(1 + \frac{a}{2}\gamma_\ell^p\right)(1 - \gamma_\ell a) \le 1 - \frac{a}{2}\gamma_\ell.$$

The proof is concluded. $\qquad\square$

| Tasks | $\bar{\epsilon} = \epsilon_{\mathsf{avg}}$ | $a_0$ | $a_1$ | `batch` |
|---|---|---|---|---|
| Gaussian Mean Estimation | see §4 | 50 | 10000 | 1 |
| Spam Email Classification | see §4 | 50 | 100000 | 32 |
| LEAF Synthetic Data (Hetero & Homo) | 0.1 | 200 | 1000 | 32 |
| LEAF Synthetic Data (Hetero & Homo) | 10.0 | 1 | 1000 | 32 |

Table 1: Parameters for the numerical experiments.

**Lemma 7.** *Consider a sequence of non-negative, non-increasing step sizes $\{\gamma_t\}_{t\geq 1}$. Let $p \in \mathbb{Z}^+$. If $\sup_{t\geq 1} \gamma_t^p/\gamma_{t+1}^p \leq 1 + \frac{\rho}{4-2\rho}$, then for any $t \geq 0$, it holds that*

$$\sum_{i=1}^{t+1} \left(1 - \frac{\rho}{2}\right)^{t+1-i} \gamma_i^p \leq \frac{4}{\rho}\gamma_{t+1}^p. \tag{57}$$

*Proof.* We observe the following chain:

$$\sum_{i=1}^{t+1} \left(1 - \frac{\rho}{2}\right)^{t+1-i} \gamma_i^p = \gamma_{t+1}^p \sum_{i=1}^{t+1} \left(1 - \frac{\rho}{2}\right)^{t+1-i} \left(\frac{\gamma_i}{\gamma_{i+1}}\right)^p \left(\frac{\gamma_{i+1}}{\gamma_{i+2}}\right)^p \cdots \left(\frac{\gamma_t}{\gamma_{t+1}}\right)^p$$

$$\leq \gamma_{t+1}^p \sum_{i=1}^{t+1} (1 - \frac{\rho}{4})^{t+1-i} \leq \frac{4}{\rho}\gamma_{t+1}^p$$

where the second last inequality is due to:

$$(1 - \rho/2)\left(\frac{\gamma_{i+1}}{\gamma_{i+2}}\right)^p \leq 1 - \frac{\rho}{4}$$

since $\sup_{k\geq 1} \gamma_{k-1}^p/\gamma_k^p \leq 1 + \frac{\rho}{4-2\rho}$. This completes the proof. $\square$

# F    Details of Numerical Experiments and Additional Results

This section provides details for the numerical experiments conducted in §4. We also describe an additional numerical experiment based on the logistic regression Example 1 on synthetic data. The latter examines the effects of heterogeneous data on the convergence rate of `DSGD-GD`.

For all our experiments, we have performed `DSGD-GD` with the step size $\gamma_t = a_0/(a_1 + t)$. Moreover, at each iteration of `DSGD-GD`, the $i$th agent draws `batch` $\geq 1$ samples from $\mathcal{D}_i(\boldsymbol{\theta}_i^t)$. The parameters $a_0 > 0, a_1 \geq 0$, `batch` $\geq 1$ used for different tasks are specified in Table 1. For both Gaussian mean estimation and `spambase` logistic regression, we use the same parameters for all settings of $\bar{\epsilon} = \epsilon_{\mathsf{avg}}$. We consider using $n = 25$ agents in all experiments, connected on a ring graph. We set the mixing matrix weights as $W_{ij} = 1/3$ for all $(i,j) \in E$, and $W_{ij} = 0$ if $(i,j) \notin E$.

**Spam Email Classification.** In Fig. 4, we provide additional results for the experiment in the main paper [cf. Fig. 3]. In particular, we compare the training loss $f(\overline{\boldsymbol{\theta}}^t; \overline{\boldsymbol{\theta}}^t)$ and training accuracy against the iteration number $t$. We also plot the gap to an *approximate* Multi-PS solution in Fig. 4 (right) for $\|\overline{\boldsymbol{\theta}}^t - \boldsymbol{\theta}^{\hat{P}S}\|^2$. Note that the Multi-PS solution compared here is only an approximation obtained by applying a similar method to repeated gradient descent in [Perdomo et al., 2020] on $\min_{\boldsymbol{\theta}} \sum_{i=1}^n f_i(\boldsymbol{\theta}; \boldsymbol{\theta})$, where we used 1000 gradient descent iterations together with an outer loop of $10^4$ deployments. Note that this process is only guaranteed to find a near-optimal solution, denoted as $\boldsymbol{\theta}^{\hat{P}S}$. Nevertheless, we observe that when the decision dependent distributions becomes more sensitive ($\epsilon_{\mathsf{avg}} = 1$), the `DSGD-GD` scheme seems unable to reach $\boldsymbol{\theta}^{\hat{P}S}$.

**Logistic Regression on `LEAF` Synthetic Data.** To study the effect of homogeneity of data distribution [cf. A6] on the convergence of `DSGD-GD`, we conduct an additional experiment based on Example 1 but on the `LEAF` synthetic data [Caldas et al., 2019]. Here, we set the sensitivity parameter at $\epsilon_i = \bar{\epsilon}$ for $i = 1, \ldots, 25$ and generate synthetic data using the framework in [Caldas et al., 2019] with the

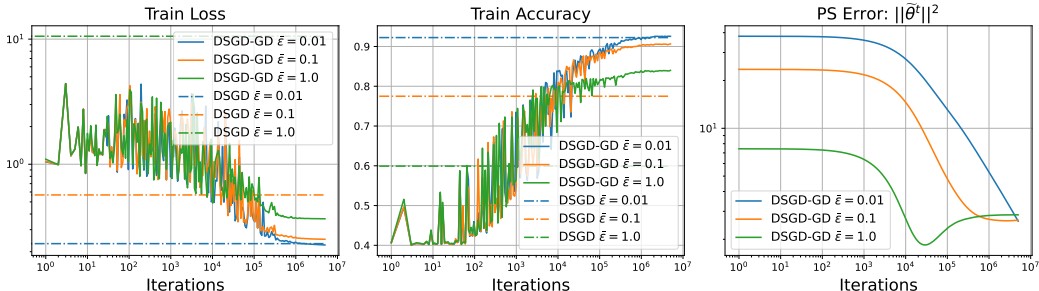

Figure 4: **Additional Results for Spam Email Classification.** (Left) Training Loss. (Middle) Training Accuracy. (Right) Approximate Gap to Multi-PS solution (see below). We also compare the non-performative optimal solution (dashed lines) on the shifted dataset.

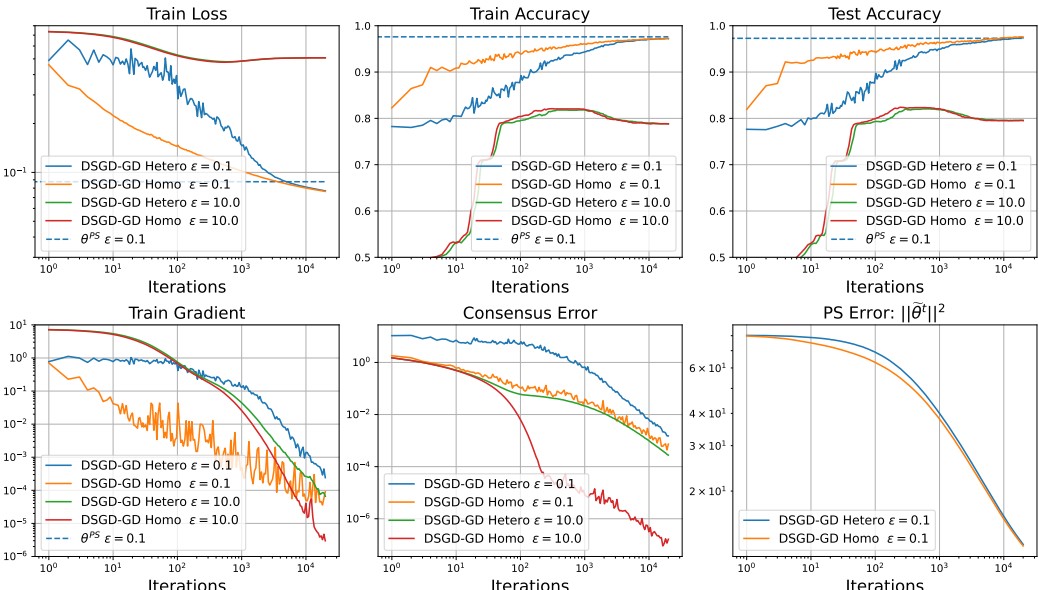

Figure 5: **Logistic Regression on `LEAF` Synthetic Data.** DSGD-GD in homogeneous and heterogeneous data distribution converge to the same Multi-PS solution.

standard deviation $\sigma = 1$ that represents the degree of heterogeneity of the dataset. Note that the framework produces $m_i = 100$ training samples with $d = 100$ features for each agent, denoted as $(\boldsymbol{X}_k^i, Y_k^i)_{k=1}^{100}$, for $i = 1, \dots, 25$ agents.

We consider two settings and describe them using the notations as in Example 1. In the `heterogeneous` data setting, the base data distribution $\mathcal{D}_i^0$ for agent $i$ is taken to be $(\boldsymbol{X}_k^i, Y_k^i)_{k=1}^{100}$ such that $\mathcal{D}_i(\boldsymbol{\theta}) \neq \mathcal{D}_j(\boldsymbol{\theta})$. In the `homogeneous` data setting, the base data distribution $\mathcal{D}_i^0$ for agent $i$ is taken to be $((\boldsymbol{X}_k^i, Y_k^i)_{k=1}^{100})_{i=1}^{25}$, i.e., the entire dataset generated from `LEAF`. Note that in this case, $\mathcal{D}_i^0 \equiv \mathcal{D}_j^0$ and thus $\mathcal{D}_i(\boldsymbol{\theta}) \equiv \mathcal{D}_j(\boldsymbol{\theta})$ for any $\boldsymbol{\theta} \in \mathbb{R}^d$ and $i, j = 1, \dots, n$ since $\epsilon_i = \epsilon_{\mathsf{avg}}$. Note that the Multi-PS solution $\boldsymbol{\theta}^{PS}$ (if exists) in both settings are unique and identical. Meanwhile, the `homogeneous` case satisfies A6 with $\varsigma = 0$, thus the `DSGD-GD` scheme applied to it is expected to converge at a faster rate than in the `heterogeneous` case.

Our numerical results are presented in Fig. 5, and we show in Table 1 the simulation parameters. Observe that with $\epsilon_{\mathsf{avg}} = 10$, the local data distributions are too sensitive and the Multi-PS solution $\boldsymbol{\theta}^{PS}$ may not exist. With $\epsilon_{\mathsf{avg}} = 0.1$, we observe that the convergence of test accuracy, training loss, etc. are faster with the `homogeneous` case initially. However, as the iteration number $t$ grows, the gap between the `homogeneous` and `heterogeneous` cases fade. This corroborates with our finite-time analysis in (18), where the fluctuation term $\sigma^2 \gamma_t / (n\widetilde{\mu})$ becomes dominant as $t \gg 1$ in all cases, yet the transient time can be shorter when $\varsigma = 0$ as predicted by (19).

# G  Extension to Time-varying Graph

This section shows how to extend our analysis for `DSGD-GD` to the setting with time-varying communication graph. Let $G^{(t)} = (V, E^{(t)})$ be a simple, undirected graph which is possibly not connected and the graph is associated with a weighted adjacency matrix $W^{(t)}$. Note that the graph $G^{(t)}$ consists of a fixed set of agents $V$ and a set of time-varying edges $E^{(t)}$.

In lieu of A4, we assume that:

**A7.** *The time-varying undirected graph sequence* $\{G^{(t)}\}_{t \geq 1} = \{(V, E^{(t)})\}_{t \geq 1}$ *is $B$-connected. Specifically, for any $t \geq 1$, there exists a positive integer $B$ such that the undirected graph $(V, E^{(t)} \cup \cdots E^{(t+B-1)})$ is connected. For any $t \geq 1$, the mixing matrix $W^{(t)} \in \mathbb{R}^{n \times n}$ satisfies:*

1. *(Topology)* $W_{ij}^{(t)} = 0$ *if* $(i, j) \notin E^{(t)}$.

2. *(Doubly stochastic)* $W^{(t)} \mathbf{1} = (W^{(t)})^\top \mathbf{1} = \mathbf{1}$.

3. *(Fast mixing) Let* $A^{(t)} := W^{(t)} - \frac{1}{n} \mathbf{1}\mathbf{1}^\top$, *there exists* $\bar{\rho} \in (0, 1]$ *such that* $\left\| A^{(t+B-1)} \cdots A^{(t)} \right\|_2 \leq 1 - \bar{\rho}$.

*The last condition can be guaranteed under the bounded communication setting, i.e., when the combined graph $(V, E^{(t)} \cup \cdots E^{(t+B)})$ is connected for any $t \geq 0$.*

**Notations.** Throughout, we denote $\Theta(m, n) := \mathbb{E}[\|\Theta_o^m\|_F^2 + \cdots + \|\Theta_o^n\|_F^2]$ and $\widetilde{\theta}(m, n) := \mathbb{E}[\|\widetilde{\theta}^m\|^2 + \cdots + \|\widetilde{\theta}^n\|^2]$, which is the aggregation of consensus error and performative stable gap in one time block whose length is $B$, respectively.

**Proof Sketch.** Below we provide a proof sketch for the convergence of `DSGD-GD` scheme when the latter is applied on a time varying graph satisfying A7. We begin by considering the extensions of Lemmas 3 and 4. As follows,

**Lemma 8 (Extension of Lemma 3).** *Fix any $\delta > 0$ and let $\epsilon_{\mathsf{avg}} \leq \frac{\mu}{(1+\delta)L}$. Under A1, A2, A3, A5 and let the step sizes satisfy $\sup_{t \geq 0} \gamma_{t+1} \leq \frac{\widetilde{\mu}}{c_2}$, the following bound holds for any $t \geq 0$,*

$$
\widetilde{\theta}(t+1, t+B) \leq (1 - \widetilde{\mu}\gamma_{t+B})^B \widetilde{\theta}(t-B+1, t) + \frac{2B\sigma^2}{n}\gamma_{t+1}^2
$$
$$
+ B\left(c_1 \frac{\gamma_{t+1}}{n} + c_2 \frac{\gamma_{t+1}^2}{n}\right)[\Theta(t-B+1, t) + \Theta(t+1, t+B)].
$$

*Proof.* Recall the inequality (22) in Lemma 3,

$$
\mathbb{E}_t \left\| \widetilde{\theta}^{t+1} \right\|^2 \leq (1 - \widetilde{\mu}\gamma_{t+1}) \left\| \widetilde{\theta}^t \right\|^2 + [c_1\gamma_{t+1} + c_2\gamma_{t+1}^2] \frac{1}{n} \left\| \Theta_o^t \right\|_F^2 + \frac{2\sigma^2}{n}\gamma_{t+1}^2. \tag{58}
$$

This implies

$$
\widetilde{\theta}(t+1, t+B) \leq (1 - \widetilde{\mu}\gamma_{t+B})\widetilde{\theta}(t, t+B-1) + \left(\frac{c_1\gamma_{t+1}}{n} + \frac{c_2\gamma_{t+1}^2}{n}\right)\Theta(t, t+B-1) + \frac{2B\sigma^2}{n}\gamma_{t+1}^2,
$$

where we have summed (58) from $t + 1$th to $t + B$th iteration and noted that the step size $\gamma_t$ is non-increasing. Applying the above inequality for $B$ times, we can link two consecutive $B$ performative stable gap $\widetilde{\theta}(t+1, t+B)$ and $\widetilde{\theta}(t-B+1, t)$ by

$$
\widetilde{\theta}(t+1, t+B) \leq (1 - \widetilde{\mu}\gamma_{t+B})^B \widetilde{\theta}(t-B+1, t) + \frac{2B\sigma^2}{n}\gamma_{t+1}^2
$$
$$
+ \left(\frac{c_1\gamma_{t+1}}{n} + \frac{c_2\gamma_{t+1}^2}{n}\right)[\Theta(t, t+B-1) + \Theta(t-1, t+B-2) + \cdots + \Theta(t-B+1, t)].
\tag{59}
$$

For the first term $\Theta(t, t+B-1)$ in the last quantity, we observe the crude bound

$$
\Theta(t, t+B-1) \leq \Theta(t+1, t+B) + \mathbb{E}\|\Theta_o^t\|_F^2 \leq \Theta(t-B+1, t) + \Theta(t+1, t+B).
$$

Following the same trick, we get another crude bound as

$$[\boldsymbol{\Theta}(t, t+B-1) + \boldsymbol{\Theta}(t-1, t+B-2) + \cdots + \boldsymbol{\Theta}(t-B+1, t)]$$
$$\leq B[\boldsymbol{\Theta}(t-B+1, t) + \boldsymbol{\Theta}(t+1, t+B)].$$

Substituting back to inequality (59) derives the final bound

$$\widetilde{\boldsymbol{\theta}}(t+1, t+B) \leq (1 - \widetilde{\mu}\gamma_{t+B})^B \widetilde{\boldsymbol{\theta}}(t-B+1, t) + \frac{2B\sigma^2}{n}\gamma_{t+1}^2$$
$$+ B\Big(c_1 \frac{\gamma_{t+1}}{n} + c_2 \frac{\gamma_{t+1}^2}{n}\Big)[\boldsymbol{\Theta}(t-B+1, t) + \boldsymbol{\Theta}(t+1, t+B)].$$

$\square$

**Lemma 9 (Extension of Lemma 4).** *Under A2–A5 and A7 and let the step sizes satisfy*

$$\sup_{t \geq 0} \gamma_{t+1} \leq \rho/\sqrt{2Bc_3},$$

*then it holds for any $t \geq 0$ that*

$$\boldsymbol{\Theta}(t+1, t+B) \leq \frac{1 - \bar{\rho}/2}{1 - Bc_3\gamma_{t-B+1}^2/\bar{\rho}}\boldsymbol{\Theta}(t-B+1, t)$$
$$+ \frac{\gamma_{t-B+1}^2}{\rho - Bc_3\gamma_{t-B+1}^2}\Big\{B^2 d_1 + d_2 B[\widetilde{\boldsymbol{\theta}}(t-B+1, t) + \widetilde{\boldsymbol{\theta}}(t+1, t+B)]\Big\},$$

(60)

*where $d_1 := 9n(\sigma^2 + \varsigma^2)$, $d_2 := 12n(\sigma^2 + \varsigma^2)$.*

*Proof.* Recall the notations (45) and observe that

$$\boldsymbol{\Theta}_o^{t+1} = \boldsymbol{\Theta}^{t+1} - \overline{\boldsymbol{\Theta}}^{t+1} = \underbrace{\Big(\boldsymbol{W}^{t+1} - \frac{1}{n}\mathbf{1}\mathbf{1}^\top\Big)}_{=\boldsymbol{A}^{t+1}}\boldsymbol{\Theta}_o^t - \gamma_{t+1}\Big(\boldsymbol{I} - \frac{1}{n}\mathbf{1}\mathbf{1}^\top\Big)\widetilde{\nabla}F^t.$$

Therefore, we can obtain the following consensus error recursion

$$\boldsymbol{\Theta}_o^{t+1} = \boldsymbol{A}^{t+1}\boldsymbol{\Theta}_o^t - \gamma_{t+1}\big(\boldsymbol{I} - (1/n)\mathbf{1}\mathbf{1}^\top\big)\widetilde{\nabla}F^t.$$

Then, we aim to link $\boldsymbol{\Theta}_o^{t+1}$ to $\boldsymbol{\Theta}_o^{t-B+1}$.

$$\boldsymbol{\Theta}_o^{t+1} = \boldsymbol{A}^{t+1}\boldsymbol{\Theta}_o^t - \gamma_t\big(\boldsymbol{I} - (1/n)\mathbf{1}\mathbf{1}^\top\big)\widetilde{\nabla}F^{t-1}$$
$$= \boldsymbol{A}^{t+1}\boldsymbol{A}^t\boldsymbol{\Theta}_o^{t-1} - \gamma_t\boldsymbol{A}^{t+1}\big(\boldsymbol{I} - (1/n)\mathbf{1}\mathbf{1}^\top\big)\widetilde{\nabla}F^{t-1} - \gamma_{t+1}\big(\boldsymbol{I} - (1/n)\mathbf{1}\mathbf{1}^\top\big)\widetilde{\nabla}F^t$$
$$\vdots$$
$$= \boldsymbol{A}^{t+1}\boldsymbol{A}^t\boldsymbol{A}^{t-1}\cdots\boldsymbol{A}^{t-B+1}\boldsymbol{\Theta}_o^{t-B+1} - \sum_{s=t-B+1}^{t}\gamma_{s+1}\boldsymbol{A}^{s+2}\big(\boldsymbol{I} - (1/n)\mathbf{1}\mathbf{1}^\top\big)\widetilde{\nabla}F^s.$$

Taking Frobenius norm on both sides and applying the Young's inequality give

$$\big\|\boldsymbol{\Theta}_o^{t+1}\big\|_F^2 \leq (1+\alpha)\big\|\boldsymbol{A}^{t+1}\boldsymbol{A}^t\boldsymbol{A}^{t-1}\cdots\boldsymbol{A}^{t-B+1}\big\|^2\big\|\boldsymbol{\Theta}_o^{t-B+1}\big\|_F^2$$
$$+ (1+\alpha^{-1})\sum_{s=t-B+1}^{t}\gamma_{s+1}^2\big\|\boldsymbol{A}^{s+2}\big\|^2\big\|\big(\boldsymbol{I} - (1/n)\mathbf{1}\mathbf{1}^\top\big)\widetilde{\nabla}F^s\big\|_F^2,$$

which holds for any $\alpha > 0$. Using A7 and setting $\alpha = \frac{\rho}{1-\rho}$, we have

$$\big\|\boldsymbol{\Theta}_o^{t+1}\big\|_F^2 \leq (1-\bar{\rho})\big\|\boldsymbol{\Theta}_o^{t-B+1}\big\|_F^2 + \sum_{s=t-B+1}^{t}\gamma_{s+1}^2\big\|\big(\boldsymbol{I} - (1/n)\mathbf{1}\mathbf{1}^\top\big)\widetilde{\nabla}F^s\big\|_F^2.$$

Similarly, we get

$$\left\|\boldsymbol{\Theta}_o^{t+2}\right\|_F^2 \le (1-\bar{\rho})\left\|\boldsymbol{\Theta}_o^{t-B+2}\right\|_F^2 + \sum_{s=t-B+2}^{t+1} \gamma_{s+1}^2 \left\|\left(\boldsymbol{I} - (1/n)\mathbf{1}\mathbf{1}^\top\right)\widetilde{\nabla}F^s\right\|_F^2$$

$$\vdots$$

$$\left\|\boldsymbol{\Theta}_o^{t+B}\right\|_F^2 \le (1-\bar{\rho})\left\|\boldsymbol{\Theta}_o^{t}\right\|_F^2 + \sum_{s=t}^{t+B-1} \gamma_{s+1}^2 \left\|\left(\boldsymbol{I} - (1/n)\mathbf{1}\mathbf{1}^\top\right)\widetilde{\nabla}F^s\right\|_F^2.$$

Adding these $B$ consensus errors together leads to

$$\boldsymbol{\Theta}(t+1, t+B) \le (1-\bar{\rho})\boldsymbol{\Theta}(t-B+1, t)$$
$$+ \frac{\gamma_{t-B+1}^2}{\rho}\left\{\sum_{s=t-B+1}^{t} \left\|\left(\boldsymbol{I} - (1/n)\mathbf{1}\mathbf{1}^\top\right)\widetilde{\nabla}F^s\right\|_F^2 + \cdots + \sum_{s=t}^{t+B} \left\|\left(\boldsymbol{I} - (1/n)\mathbf{1}\mathbf{1}^\top\right)\widetilde{\nabla}F^s\right\|_F^2\right\}. \tag{61}$$

Using the inequality (49) in the proof of Lemma 4, we get

$$\mathbb{E}_s\left\|\left(\boldsymbol{I} - (1/n)\mathbf{1}\mathbf{1}^\top\right)\widetilde{\nabla}F^s\right\|_F^2 \le d_1 + d_2\left\|\widetilde{\boldsymbol{\theta}}^s\right\|^2 + c_3\left\|\boldsymbol{\Theta}_o^s\right\|_F^2,$$

where $d_1 := 9n(\sigma^2 + \varsigma^2)$, $d_2 := 12n(\sigma^2 + \varsigma^2)$ and $c_3 = 12\sigma^2 + 18L^2(1 + \epsilon_{\max})^2$. Then, we have

$$\sum_{s=t-B+1}^{t} \mathbb{E}\left\|\left(\boldsymbol{I} - (1/n)\mathbf{1}\mathbf{1}^\top\right)\widetilde{\nabla}F^s\right\|_F^2 \le \sum_{s=t-B+1}^{t} \mathbb{E}\left[d_1 + d_2\|\widetilde{\boldsymbol{\theta}}^s\|^2 + c_3\left\|\boldsymbol{\Theta}_o^s\right\|_F^2\right]$$

$$= Bd_1 + d_2\sum_{s=t-B+1}^{t} \mathbb{E}\|\widetilde{\boldsymbol{\theta}}^t\|^2 + c_3\sum_{s=t-B+1}^{t} \mathbb{E}\left\|\boldsymbol{\Theta}_o^s\right\|_F^2.$$

Substituting back to (61) give us

$$\boldsymbol{\Theta}(t+1, t+B) \le (1-\bar{\rho})\boldsymbol{\Theta}(t-B+1, t) + \frac{\gamma_{t-B+1}^2}{\rho}\Big\{B^2 d_1 + d_2[\widetilde{\boldsymbol{\theta}}(t-B+1, t) + \cdots + \widetilde{\boldsymbol{\theta}}(t, t+B)]$$

$$+ c_3[\boldsymbol{\Theta}(t-B+1, t) + \cdots + \boldsymbol{\Theta}(t, t+B)]\Big\}.$$

The above can be simplified to

$$\boldsymbol{\Theta}(t+1, t+B) \le (1-\bar{\rho})\boldsymbol{\Theta}(t-B+1, t) + \frac{\gamma_{t-B+1}^2}{\rho}\Big\{B^2 d_1 + d_2 B[\widetilde{\boldsymbol{\theta}}(t+1, t+B) + \widetilde{\boldsymbol{\theta}}(t, t+B)]$$

$$+ c_3 B[\boldsymbol{\Theta}(t-B+1, t) + \boldsymbol{\Theta}(t+1, t+B)]\Big\}.$$

Setting $\sup_{k\ge 1}\gamma_k \le \frac{\bar{\rho}}{\sqrt{2c_3 B}}$ and rearranging terms give us

$$\boldsymbol{\Theta}(t+1, t+B) \le \frac{1-\bar{\rho}/2}{1 - Bc_3\gamma_{t-B+1}^2/\bar{\rho}}\boldsymbol{\Theta}(t-B+1, t)$$

$$+ \frac{\gamma_{t-B+1}^2}{\rho - Bc_3\gamma_{t-B+1}^2}\Big\{B^2 d_1 + d_2 B[\widetilde{\boldsymbol{\theta}}(t-B+1, t) + \widetilde{\boldsymbol{\theta}}(t+1, t+B)]\Big\},$$

which gives us desired upper bound for $\boldsymbol{\Theta}(t+1, t+B)$. $\qquad\square$

**Convergence of $\overline{\boldsymbol{\theta}}^t$ to $\theta^{PS}$ with Time varying graph.** We conclude our proof sketch through analyzing the following Lyapunov function. For any $t \ge 0$, we define:

$$\mathcal{L}_{t+1}^{t+B} := \widetilde{\boldsymbol{\theta}}(t+1, t+B) + \boldsymbol{\Theta}(t+1, t+B) \ge 0.$$

Combing Lemma 8 and 9 leads to

$$\left(1 - Bc_1 \frac{\gamma_{t+1}}{n} - Bc_2 \frac{\gamma_{t+1}^2}{n}\right) \Theta(t+1, t+B) + \left(1 - \frac{d_2 B \gamma_{t-B+1}^2}{\rho - Bc_3 \gamma_{t-B+1}^2}\right) \widetilde{\boldsymbol{\theta}}(t+1, t+B)$$

$$\leq \left(\frac{1 - \bar{\rho}/2}{1 - Bc_3 \gamma_{t-B+1}^2/\bar{\rho}} + Bc_1 \frac{\gamma_{t+1}}{n} + Bc_2 \frac{\gamma_{t+1}^2}{n}\right) \Theta(t-B+1, t) \tag{62}$$

$$+ \left((1 - \widetilde{\mu}\gamma_{t+B})^B + \frac{d_2 B \gamma_{t-B+1}^2}{\rho - Bc_3 \gamma_{t-B+1}^2}\right) \widetilde{\boldsymbol{\theta}}(t-B+1, t) + \frac{B^2 d_1 \gamma_{t-B+1}^2}{\rho - Bc_3 \gamma_{t-B+1}^2} + \frac{2B\sigma^2}{n}\gamma_{t+1}^2.$$

We focus on the l.h.s. of above inequality. If the step size satisfies

$$\sup_{k \geq 1} \gamma_k \leq \min\left\{\frac{c_1}{c_2}, \sqrt{\frac{\bar{\rho}}{2Bc_3}}, \frac{\bar{\rho}c_1}{n}\right\}$$

then, the l.h.s. of (62) can be lower bounded by

$$\text{l.h.s. of (62)} \geq \left(1 - 2Bc_1 \frac{\gamma_{t+1}}{n}\right)\left[\Theta(t+1, t+B) + \widetilde{\boldsymbol{\theta}}(t+1, t+B)\right].$$

Next, we consider the r.h.s. of (62). Suppose that $\sup_{k \geq 1} \gamma_k \leq \sqrt{\frac{\rho}{(4-\bar{\rho})Bc_3}}$, it holds

$$\frac{1 - \bar{\rho}/2}{1 - Bc_3 \gamma_{t-B+1}^2/\bar{\rho}} \leq 1 - \bar{\rho}/4, \qquad \frac{B^2 d_1 \gamma_{t-B+1}^2}{\rho - Bc_3 \gamma_{t-B+1}^2} \leq \frac{2B^2 d_1}{\rho}\gamma_{t-B+1}^2.$$

If step size also satisfies

$$\sup_{k \geq 1} \gamma_k \leq \min\left\{\frac{1}{\sqrt{b}}, \frac{\widetilde{\mu}\bar{\rho}}{2^{2B+1}d_2 B}\right\},$$

where $b$ such that $\gamma_k^2/\gamma_{k+1}^2 \leq 1 + b\gamma_{k+1}^2$, then it holds:

r.h.s. of (62)

$$\leq \left(1 - \frac{\bar{\rho}}{4} + 2Bc_1 \frac{\gamma_{t+1}}{n}\right) \Theta(t-B+1, t) + \left(1 - \frac{\widetilde{\mu}\gamma_{t+B}}{2}\right) \widetilde{\boldsymbol{\theta}}(t-B+1, t)$$

$$+ \frac{2B^2 d_1}{\rho}\gamma_{t-B+1}^2 + \frac{2B\sigma^2}{n}\gamma_{t+1}^2.$$

Combining the above inequalities lead to:

$$\left(1 - 2Bc_1 \frac{\gamma_{t+1}}{n}\right)\left[\Theta(t+1, t+B) + \widetilde{\boldsymbol{\theta}}(t+1, t+B)\right] \leq \left(1 - \frac{\bar{\rho}}{4} + 2Bc_1 \frac{\gamma_{t+1}}{n}\right)\Theta(t-B+1, t)$$

$$+ \left(1 - \frac{\widetilde{\mu}\gamma_{t+B}}{2}\right) \widetilde{\boldsymbol{\theta}}(t-B+1, t) + \frac{2B^2 d_1}{\rho}\gamma_{t-B+1}^2 + \frac{2B\sigma^2}{n}\gamma_{t+1}^2.$$

If the step size satisfying

$$\sup_{k \geq 1} \gamma_k \leq \frac{\bar{\rho}}{8Bc_1/n + 2\widetilde{\mu}},$$

then the main recursion can be simplified as

$$\left(1 - 2Bc_1 \frac{\gamma_{t+1}}{n}\right)\left[\Theta(t+1, t+B) + \widetilde{\boldsymbol{\theta}}(t+1, t+B)\right]$$

$$\leq \left(1 - \frac{\widetilde{\mu}\gamma_{t+B}}{2}\right)\left[\Theta(t-B+1, t) + \widetilde{\boldsymbol{\theta}}(t-B+1, t)\right] + \frac{2B^2 d_1}{\rho}\gamma_{t-B+1}^2 + \frac{2B\sigma^2}{n}\gamma_{t+1}^2.$$

Dividing $\left(1 - 2Bc_1 \frac{\gamma_{t+1}}{n}\right)$ for the both sides, we obtain that

$$\mathcal{L}_{t+1}^{t+B} \leq \frac{(1 - \widetilde{\mu}\gamma_{t+B}/2)}{1 - 2Bc_1\gamma_{t+1}/n}\mathcal{L}_{t-B+1}^t + \left(\frac{2B^2 d_1}{\rho} + \frac{2B^2\sigma^2}{n}\right)\frac{\gamma_{t-B+1}^2}{(1 - 2Bc_1\gamma_{t+1}/n)}.$$

Observe that with sufficiently small step size, the above recursion can be simplified to give a similar form as (25). Solving the recursion then lead to $\mathcal{L}_{t+1}^{t+B} = \mathcal{O}(\gamma_{t-B+1})$ and the convergence of $\widetilde{\boldsymbol{\theta}}(t+1, t+B) \to 0$.

Lastly, we remark that the above analysis only gives a crude bound to the convergence of `DSGD-GD` in the time varying graph setting. It is possible to give tighter bounds through further optimizing the constants in the above analysis.

# H  Extension to Local Distributions Influenced by All Agents

This section outlines how to extend our analysis to the scenario when the local distributions $\mathcal{D}_i(\cdot)$ are simultaneously influenced by other agents in the network similar to the competitive Multi-PfD considered by [Narang et al., 2022, Piliouras and Yu, 2022].

We define the concatenated decision vector $\boldsymbol{\vartheta} := (\boldsymbol{\theta}_1, \ldots, \boldsymbol{\theta}_n) \in \mathbb{R}^{nd}$ and state the modified consensus Multi-PfD problem (1) as follows

$$\min_{\boldsymbol{\theta}_i \in \mathbb{R}^d, i=1,\ldots,n} \frac{1}{n} \sum_{i=1}^n \mathbb{E}_{Z_i \sim \mathcal{D}_i(\boldsymbol{\vartheta})} \big[ \ell(\boldsymbol{\theta}_i; Z_i) \big] \ \text{ s.t. } \ \boldsymbol{\theta}_i = \boldsymbol{\theta}_j, \ \forall \ (i,j) \in E. \tag{63}$$

With a slight abuse of notation, we also define $f_i(\boldsymbol{\theta}; \boldsymbol{\vartheta}) := \mathbb{E}_{Z_i \sim \mathcal{D}_i(\boldsymbol{\vartheta})} \big[ \ell(\boldsymbol{\theta}_i; Z_i) \big]$.

Following [Narang et al., 2022], we consider the following modification to A3:

**A8.** *For any $i = 1, \ldots, n$, there exists a constant $\epsilon_i > 0$ such that*

$$\mathcal{W}_1(\mathcal{D}_i(\boldsymbol{\vartheta}), \mathcal{D}_i(\boldsymbol{\vartheta}')) \leq \epsilon_i \left\| \boldsymbol{\vartheta} - \boldsymbol{\vartheta}' \right\|, \ \forall \ \boldsymbol{\vartheta}', \boldsymbol{\vartheta} \in \mathbb{R}^{nd}, \tag{64}$$

*where $\mathcal{W}_1(\mathcal{D}, \mathcal{D}')$ denotes the Wasserstein-1 distance between the distributions $\mathcal{D}, \mathcal{D}'$.*

Specifically, we notice that if $\boldsymbol{\vartheta}$ satisfies the consensus constraint, i.e., $\boldsymbol{\vartheta} = \mathbf{1}_n \otimes \boldsymbol{\theta} = (\boldsymbol{\theta}, \ldots, \boldsymbol{\theta})$, then A8 is equivalent to A3 with the latter's sensitivity parameter given by $\epsilon_i' = \sqrt{n}\epsilon_i$ since $\| \mathbf{1}_n \otimes \boldsymbol{\theta} - \mathbf{1}_n \otimes \boldsymbol{\theta}' \| = \sqrt{n}\|\boldsymbol{\theta} - \boldsymbol{\theta}'\|$. This observation immediately leads to the following corollary of Proposition 1:

**Corollary 1.** *Under A1, A2, A8. Define the map $\mathcal{M} : \mathbb{R}^d \to \mathbb{R}^d$*

$$\mathcal{M}(\boldsymbol{\theta}) = \arg \min_{\boldsymbol{\theta}' \in \mathbb{R}^d} \frac{1}{n} \sum_{i=1}^n f_i(\boldsymbol{\theta}'; \mathbf{1}_n \otimes \boldsymbol{\theta}) \tag{65}$$

*If $\sqrt{n}\epsilon_{\mathsf{avg}} < \mu/L$, then the map $\mathcal{M}(\boldsymbol{\theta})$ is a contraction with the unique fixed point $\boldsymbol{\theta}^{PS} = \mathcal{M}(\boldsymbol{\theta}^{PS})$. If $\sqrt{n}\epsilon_{\mathsf{avg}} \geq \mu/L$, then there exists an instance of (11) where $\lim_{T \to \infty} \left\| \mathcal{M}^T(\boldsymbol{\theta}) \right\| = \infty$.*

The proof is attained by simply observing that if $\boldsymbol{\theta}_i = \boldsymbol{\theta}_j$ (as constrained by (65) (and (63)), then A8 is equivalent to A3 with $\epsilon_i' = \sqrt{n}\epsilon_i$.

**Comparison to [Narang et al., 2022].** Notice that in [Narang et al., 2022], the existence of a performative stable equilibrium requires $\sqrt{\sum_{i=1}^n \epsilon_i^2} < \mu/L$. Meanwhile, Corollary 1 requires $(1/\sqrt{n}) \sum_{i=1}^n \epsilon_i < \mu/L$. Due to norm equivalence, we have

$$(1/\sqrt{n}) \sum_{i=1}^n \epsilon_i \leq \sqrt{\sum_{i=1}^n \epsilon_i^2}.$$

Thus, the consensus constrained performative stable solution in cooperative Multi-PfD will be attainable under a more relaxed condition than the competitive Multi-PfD.

`DSGD-GD` **Algorithm for** (63). The extension of Theorem 1 to (63) via the `DSGD-GD` algorithm is more involved and thus the details are skipped in this brief discussion. However, it remains straightforward to extend the analysis through a careful modification of Lemma 3 with A8. In particular, one only needs to pay attention to the use of A8 in (37) and (40) for the proof.

**Remarks.** We emphasize that as explained in the main paper, the original scenario considered by (1) and A3 is relevant to the decentralized learning scenario of the current paper. It captures the effects of 'geographical' barriers where the population of users are not simultaneously influenced by all agents. Nevertheless, a future direction is to study the Multi-PfD problem (cooperative or competitive) where users can be influenced by the decisions from a *few* neighboring agents.