# OpenReview forum: "Multi-agent Performative Prediction with Greedy Deployment and Consensus Seeking Agents"
_NeurIPS.cc/2022/Conference — NeurIPS 2022 Accept_

### Official Review · Reviewer_JuZu · 2022-07-07

**Rating:** 4
**Confidence:** 1
**Soundness:** 2 fair
**Presentation:** 3 good
**Contribution:** 2 fair

**Summary:**

This paper formulate Multi-PfD as a decentralized optimization problem and admit a multiagent solution. Compared to the single-agent case, enforcing consensus leads to a laxer condition for the existence of Multi-PS solution with respect to the distributions’ sensitivity. This paper study the DSGD-GD scheme and analyze its convergence towards the Multi-PS solution, and show that the scheme is convergent under the same sufficient condition for existence of Multi-PS solution

**Questions:**

Can the method be used on reinforcement learning to change the unique convergence problem of reinforcement learning algorithm?

**Strengths And Weaknesses:**

originality:
Compared to multi-agent reinforcement learning algorithms, the proposed DSGD-DG method can converge to a unique Multi-PS solution.

quality, clarity:
The article is well organized.


significance:
The algorithm proposed in this paper has practical significance. In the dynamic classification task, it can dynamically adapt to some changes in the environment. Such as spam classification tasks. The algorithm was also experimented on this task, and achieved good results on the spambase-based multi-agent spam classification task.

---

> ### Author Response · Authors · 2022-08-02
> **Reply to Reviewer JuZu**
>
> We thank the reviewer for his/her positive review of our paper and recognizing the practical importance of our results. Below, we address your question on:
>
> > Can the method be used on reinforcement learning to change the unique convergence problem of reinforcement learning algorithm?
>
> While many RL problems can be rewritten as (1) such as the policy optimization problem, it will be challenging to apply our results on the convergence to a unique point for these problems. Notably, our results require assumption A1 on strong convexity of the loss function with respect to the decision vector. In the case of policy optimization, this requires the loss function to be strongly convex with respect to the policy parameter. The latter condition can be hard to verify in practice. On the other hand, these assumptions are relevant to Multi-PfD problems such as with applications to decentralized learning when the optimization solution may influence the outcome they aim to predict. For instance, it is relevant to supervised learning problems with a linear model.
>
> For the contributions, in fact our paper proves new theoretical results for a fundamental issue on the Multi-PfD problem’s stability that holds promises to deepen the community’s understanding on practical ML systems. Precisely, we show that cooperation between the agents leads to a more robust Multi-PfD problem (Proposition 1). Our finding has been supported by algorithm analysis (Theorem 1) and numerical experiments (Sec. 4).
>
> We would be happy to discuss with the reviewer should there be any other concerns/questions. We would kindly ask the reviewer to consider raising his/her score if your concerns are addressed. Thank you very much!

---

> ### Author Response · Authors · 2022-08-06
> **Reply to Reviewer JuZu**
>
> Dear reviewer JuZu,
>
> We notice that the rating of your review has been changed from **5** to **4** after the author's response. However, we do not see any update in the context of your review. We are wondering about the reasons that led to the change of your mind. We would be happy to discuss if we can address your new concerns.
>
> Authors of Paper 9016.

---

### Official Review · Reviewer_N8Lb · 2022-07-10

**Rating:** 6
**Confidence:** 3
**Soundness:** 3 good
**Presentation:** 4 excellent
**Contribution:** 2 fair

**Summary:**

This paper studies a multi-agent performative prediction problem, where the agents want to learn an identical local decision parameter to minimize the average of all local loss functions. The main challenge is that the local data distribution, which is an input to the local loss function, will change as the agents update the local decision vectors. Before this work, similar performative prediction problems have been studied in single-agent settings and competitive multi-agent settings. In contrast, this work studies a cooperative multi-agent setting where all agents are willing to share local decision vectors and want to reach a consensus. The authors propose a decentralized learning scheme and provide a finite-time convergence bound. They also conduct numerical simulations on synthetic data to validate the theoretical results.

**Questions:**

As discussed in ``Weakness’’, I am concerned about the motivation behind the proposed problem setting in this work. The authors presented three examples in this paper: Example 1, multi-agent Gaussian mean estimation, and Email Spam Classification, but it looks like they are artificially constructed to fit in the Multi-PfD setting rather than coming from real-world applications. In rebuttal, I hope the authors can discuss why some core assumptions in the setting (e.g., the consensus and local communication with neighbors) are critical in these application examples.

**Limitations:**

Yes, the authors discussed the limitations of this work in the last paragraph. And I don’t see any negative social impact of this work.

**Strengths And Weaknesses:**

Strength: The paper is well-written and easy to follow, and the broad topic of performative predictive is interesting for the ML community. The authors explained the intuition behind the problem setting, the algorithm, and the main theoretical results very well. They also did a detailed comparison with related works to emphasize the novelties and challenges of this work.

Weakness: The multi-agent performative prediction problem setting proposed in this work is a novel and natural extension of the single agent setting, but the motivation is not sufficient in the following aspects: Since the local data distribution $D_i$ only depends on the local decision vector $\theta_i$, it seems unnatural to require all agents to reach a consensus on their local decision vectors. Besides, the graph $G$ of agents is only used as a communication graph. Compared with other works where the local data distribution $D_i$  can depend on non-local decision vectors, the multi-agent problem considered here looks more like a localized problem. More application examples are desired to motivate the setting considered in this work.

---

> ### Author Response · Authors · 2022-08-02
> **Reply to Reviewer N8Lb - Part 1/2**
>
> We thank the reviewer for his/her careful review and recognizing the importance of our new results on the cooperative Multi-PfD problem. It seems that the reviewer is mainly concerned with the motivation behind the application examples. We hope the following responses can clear your doubts.
>
> We first clarify about the application background. Our work is directly motivated by the wide-spread applications of decentralized algorithms to machine learning, which leverages the large amount of data at cooperating agents (could be heterogeneous due to geographical separation of agents); see [Ram et al., 2011, Lian et al., 2017] and related references. Encouragingly, we recently see that decentralized learning is applied in real world scenarios such as clinical data [Warnat-Herresthal et al., 2021], financial data [Li and Wang, 2019], etc. As pointed out by [Hardt et al., 2016, Perdomo et al., 2020], real world applications of ML systems may involve user data that can be influenced by the optimization solution leading to distribution shifts. The above premises motivated us to study the cooperative Multi-PfD framework. In view of the above, we provide novel analysis showing that the consensus requirement in decentralized learning leads to a more stable optimization compared to a single agent setup (Proposition 1). We prove that simple DSGD suffices to find a stable solution (Theorem 1) with only marginal loss in convergence rate. Below, we address your specific comments:
>
> > Since the local data distribution $D_i$ only depends on the local decision vector $\theta_i$, it seems unnatural to require all agents to reach a consensus on their local decision vectors.
>
> Thank you for the question. We believe that under the decentralized learning setting, it is natural for $D_i$ to depend on local decision $\theta_i$ only, while requiring agents to reach consensus (but only ***eventually***).
>
> First, in decentralized learning, the data under consideration are generated from groups who are separated from each other. For example with clinical data, each agent represents a hospital, and the local data represents the patients of each hospital. Naturally, the decisions made by the “local” hospital only influences its local data. We model such an influence structure with that ${\cal D}_i$ is dependent on $\theta_i$ only. Second, we note that the consensus constraint is motivated by the decentralized learning application itself, e.g., as in [Lian et al., 2017, Warnat-Herresthal et al., 2021, Li and Wang, 2019]. It is enforced so to promote properties such as better generalization performance, or is simply required by the application that requires making a consensual decision. Note that consensus is only required for the converged solution, instead of intermediate ones. In fact, our analysis (Theorem 1) is precisely about studying the dynamical behavior when the agents interact with their own performative users, while attempting to reach a consensual solution.
>
> Lastly, our study can be extended to when ${\cal D}_i$ depends on $\theta_1,...,\theta_n$, i.e., the decision vectors of all agents. This involves modifying A3 and refining the bound in Lemma 3, and it does not affect our main conclusion. We have done so in the revised Appendix H to illustrate this point. That said, we find the original setup of our paper to be more relevant to decentralized learning.
>
> > Besides, the graph $G$ of agents is only used as a communication graph.
>
> The reviewer has made a correct observation. However, $G$ is simply a part of the computation architecture in our setup. As decentralized algorithms do not require a central server, each agent is responsible for the local calculations and only communicates with neighboring agents. It is thus common to impose a communication graph to model the limitation in information exchanges pattern between agents.
>
> > Compared with other works where the local data distribution $D_i$ can depend on non-local decision vectors, the multi-agent problem considered here looks more like a localized problem.
>
> Thank you for the question. Again, the reviewer has made a correct observation. In fact, if we disconnect the $i$th agent, then it leads to a localized problem as its accompanying data ${\cal D}_i(\theta_i)$ only depends on $\theta_i$. However, doing so reduces the problem to single agent performative prediction as in [Perdomo et al., 2020] and loses the desirable properties such as having a stabler Multi-PfD problem (Proposition 1) of cooperation. To this end, we reiterate that a key message of our paper is to show that practicing cooperation in Multi-PfD is beneficial.
>
> > More application examples are desired to motivate the setting considered in this work.
>
> Thank you for the suggestion. In the main text of the final version, we will incorporate the examples mentioned in this response to support the setting considered in this work.

---

> > ### Author Response · Authors · 2022-08-02
> > **Reply to Reviewer N8Lb - Part 2/2**
> >
> > > The authors presented three examples in this paper: Example 1, multi-agent Gaussian mean estimation, and Email Spam Classification, but it looks like they are artificially constructed to fit in the Multi-PfD setting rather than coming from real-world applications.
> >
> > We emphasize that our examples were not artificially constructed to fit the Multi-PfD setting. Instead, Example 1 and the spam email filter example are modified from standard decentralized learning applications in [Lian et al., 2017, Warnat-Herresthal et al., 2021, Li and Wang, 2019]. For Gaussian mean estimation, it is related to a cooperative quadratic game in a similar vein as described in [Sec. 7.2, Narang et al., 2022]:
> >
> > + Consider $n$ branches of the same company serving different regions who aim to agree on the same price for a product. The branches wish to minimize the total negated revenue:
> > $
> > \frac{1}{n}\sum_{i=1}^{n}\frac{1}{2}z_i^\top\theta_{i} + \frac{\lambda}{2} \| \theta_i \|^2,
> > $
> > where $\lambda_i > 0$. To model influences of agent’s decision on the data [Narang et al., 2022], we have $z_i = \xi_i + A_i \theta_i$ such that $\xi_i \sim {\cal D}_i$. Observe that with ${\cal D}_i = {\cal N}(0,1)$ and $A_i = \epsilon_i$, we recover the Gaussian mean estimation problem.
> >
> > Admittedly when it comes to modeling distribution shifts in numerical experiments, our examples use a somewhat artificial setup with users applying a quadratic utility that results in a linear distribution shift. However, we note that it is a challenging open problem related to behavioral modeling in finding a realistic setup, and is out of the scope for our paper. Nevertheless, we note that such quadratic utility setups are commonly used in the performative prediction literature, e.g., [Perdomo et al., 2020] and its related references, and our theoretical analysis based on A3 applies to the general setups beyond quadratic utility.
> >
> > > I hope the authors can discuss why some core assumptions in the setting (e.g., the consensus and local communication with neighbors) are critical in these application examples.
> >
> > We have explained in the response above that the core modeling assumptions, such as consensus and local communication, are consequences of the decentralized learning applications.
> >
> > Again, we would like to thank the reviewer for his/her careful evaluation of our paper. We would be happy to discuss with the reviewer should there be any other concerns/questions. We would kindly ask the reviewer to consider raising his/her score if your concerns are addressed. Thank you very much!
> >
> > **References**
> >
> > S. Warnat-Herresthal et al., Swarm learning for decentralized and confidential clinical machine learning, Nature, 2021.
> >
> > Li, D. and Wang, J.. Fedmd: Heterogeneous federated learning via model distillation. arXiv:1910.03581, 2019.

---

> > > ### Comment · Reviewer_N8Lb · 2022-08-08
> > > **Thanks for the detailed response!**
> > >
> > > I want to thank the authors for providing the detailed response. The additional results in Appendix H are helpful to address my concern on the assumptions. However, as for the motivation, I still have some concern about the motivation for reaching the consensus and assuming all agents are willing to cooperate. Since the data distribution are assumed to be local, it seems weird to me that a global consensus is desired because it can "generalize" as the authors mentioned in the answer to the first question.

---

> > > > ### Author Response · Authors · 2022-08-08
> > > > **Reply to Reviewer N8Lb**
> > > >
> > > > We thank reviewer N8Lb for the prompt reply. It seems that the reviewer is still concerned about the motivation of our setting involving (A) local distribution and (B) desire to reach consensus. We are happy to discuss it as follows.
> > > >
> > > > Regardless of global or local distribution, there is a desire to achieve global consensus as motivated by a number of works in decentralized learning and federated learning literature. In specific, despite some agents may be simply “asked” to observe consensus, adopting a consensus design can bring benefits such as better out-of-sample performance, the ability to utilize data observed at all agents, improved robustness to node failure and local distribution shifts (e.g., performativity) that may lead to divergence, as demonstrated in this paper, etc. For example, in clinical data, each agent represents a hospital who wishes to learn about the treatment of a certain medical condition but has no previous experience (i.e., existing samples) in its local database/distribution, in this case the hospital will be motivated to cooperate with other hospitals with the desired experience (and vice versa). Notice that clinical data are privacy sensitive that shall not be shared directly. Decentralized algorithms such as DSGD are acceptable candidates here as it only requires sharing the model $\theta$ among neighboring agents.
> > > >
> > > > Besides the above example, we believe that in many applications, there is also a strong desire for agents to take on cooperative behavior. We would be glad to further discuss with the reviewer on this matter.

---

### Official Review · Reviewer_a5Cf · 2022-07-11

**Rating:** 4
**Confidence:** 3
**Soundness:** 2 fair
**Presentation:** 3 good
**Contribution:** 2 fair

**Summary:**

This paper consider the problem of multi-agent performative prediction (Multi-PfD), and formulate Multi-PfD as a decentralized optimization problem. The authors first prove the necessary and sufficient condition for the Multi-PfD problem to admit a unique multi-agent performative stable (Multi-PS) solution, show that enforcing consensus leads to a laxer condition for existence of Multi-PS solution with respect to the distributions’ sensitivities, compared to the single agent case. Then, they study a decentralized extension to the greedy deployment scheme [Mendler-Dünner et al., 2020], called the DSGD-GD scheme. They show that DSGD-GD converges to the Multi-PS solution and analyze its non-asymptotic convergence rate.




**Questions:**

Please clearly state the status of area of multi-agent performative prediction (Multi-PfD) and demonstrate your contribution positions. From the optimization side, I don't see too much novelty of the paper.

**Limitations:**

the authors adequately addressed the limitations and potential negative societal impact of their work

**Strengths And Weaknesses:**

Pros: this paper provides the study and analysis of Multi-PfD with consensus seeking agents via a practical DSGD-GD scheme. The results indicate that when agents are consensus seeking, Multi-PfD admits a performative stable solution with laxer condition and DSGD-GD achieves linear speedup.

Cons: In terms of optimization, the proposed methods are not novel, especially with limitations such as the requirement of synchronous updates among agents, strongly convex loss, etc. I would say the major contribution will be on the Multi-PfD problem formulation side, which I am not quite familiar with.

---

> ### Author Response · Authors · 2022-08-02
> **Reply to Reviewer a5Cf - Part 1/2**
>
> We thank the reviewer for his/her review and recognizing our efforts to deepen understanding of the cooperative Multi-PfD framework. It seems that the reviewer is confused about the contributions and positioning of our paper. We hope that our following responses can clear your doubts.
>
> > In terms of optimization, the proposed methods are not novel, especially with limitations such as the requirement of synchronous updates among agents, strongly convex loss, etc.
>
> Thanks for bringing this up. We have partially addressed this limitation through analyzing DSGD on ***time varying graphs*** in Appendix G. Here, we extended our analysis to a sequence of B-connected communication graphs, where the communication links between agents can be intermittent. Appendix G shows that DSGD still converges to the multi-agent performative stable point at the rate of $O(\gamma_{t-B+1})$.
>
> As for the case of non-strongly convex loss, we note that only a few recent works have partial results on the matter [Li and Wai, 2022], [Roy et al., 2022] for single agent performative prediction, where the convergence to a near-stationary point is established. For instance, [Li and Wai, 2022] achieves this result by treating the greedy-deployment scheme (which is akin to DSGD-GD) as a biased SA algorithm. For Multi-PfD, we anticipate that a similar proof will work, yet it will fall short at yielding just convergence to a near-stationary point, which is a weaker form of convergence than in our Theorem 1.
>
> > I would say the major contribution will be on the Multi-PfD problem formulation side, which I am not quite familiar with.
>
> The reviewer is correct that our contributions lie in the analysis surrounding the Multi-PfD framework with cooperating agents. The main feature of this framework is that it models a natural phenomena where the optimization solution influences the data/outcome they aim to predict. We set our problem under the background of the widespread applications of decentralized learning and related settings of federated learning to real world problems. Overall, Multi-PfD considers a scenario where local agents seek a common decision vector that fits the data of all, while the data observed can be influenced by the decision vector itself. Note that prior works only considered either a single agent setting or a competitive Multi-PfD setting.
>
> As such a **first** study on cooperative Multi-PfD, we believe that it is important to analyze how existing algorithms (e.g., DSGD) perform prior to proposing anything new. Specifically, the agents are modeled as being unaware of any potential distribution shifts. We provided an affirmative result (Theorem 1) by showing that DSGD still converges to a unique multi-agent performative stable solution. Moreover, the algorithm-independent result in Proposition 1 shows that the consensus formulation leads to a more relaxed requirement for existence/uniqueness of performative stable solution than its counterpart in the single agent setting. This shows yet another benefit of cooperation. In particular, the presence of other agents can help with reaching a performative stable solution, which is commonly sought with performative prediction.

---

> > ### Author Response · Authors · 2022-08-02
> > **Reply to Reviewer a5Cf - Part 2/2**
> >
> > > Please clearly state the status of area of Multi-PfD and demonstrate your contribution positions. From the optimization side, I don't see too much novelty of the paper.
> >
> > Our paper has made novel contributions in at least three areas:
> >
> > **Multiagent Optimization**: We advance the theory of decentralized optimization with the **first** results on the effects of distribution shift where the optimization solution influences the data it relies on. Note it is unknown in the literature if a distribution shift can destabilize the decentralized algorithm or not. As the first study, we focus on the DSGD algorithm. Here, Theorem 1 shows that DSGD still converges under the fundamental stability condition analyzed by our Proposition 1, which is also a new result. The distribution shift effects are captured by the constant $\tilde{\mu} = \mu - \epsilon_{avg} L$ where $\epsilon_{avg}$ is the averaged distribution sensitivity. As the convergence bound grows with $1/\tilde{\mu}$, we see that the convergence rate of DSGD may decrease with more sensitive distributions. Besides, the effects of distribution shift on decentralized stochastic optimization is an interesting problem. As a future direction, we believe the extension of our analysis to sophisticated algorithms like D-GET [Sun et al., 2020], EXTRA [Shi et al., 2015] can lead to useful observations regarding the robustness of these algorithms.
> >
> > **Performative Prediction**: Along the lines of [Perdomo et al, 2020] and related literature, recent works studied the convergence behavior of SGD with distribution shift under IID [Mendler et al., 2020] or Markovian data [Li and Wai, 2022]. They focus on a single agent setting. In contrast, we consider multiple cooperating agents. In this way, we contribute to the literature with the first study on cooperative Multi-PfD. One of our key findings is Proposition 1 which shows that such problem setup (independent of the algorithm used) entails a relaxed condition for guaranteeing the existence/uniqueness of the multi-agent analog of performative stable solution studied by [Perdomo et al., 2020]. Such a result comes as the benefit of cooperation between agents, and is supplemented by algorithm analysis (Theorem 1) and numerical experiments in Sec. 4.
> >
> > **Multi-agent Performative Prediction**: Recent works considered variants of Multi-PfD [Narang et al., 2022, Piliouras and Yu, 2022] from a game theoretical perspective. They differ from our paper in the sense that they considered a competitive setting and the coupling between agents is established through data that are influenced by the decisions of *all* agents (Fig. 1). Compared to our setting with cooperative agents, such a setup with data that can be universally influenced by all ignores the possibility that the population responsible for the data can be separated “geographically”. In light of this, our work offers a new perspective for investigating the Multi-PfD problem. Notice that Appendix H of the revised paper studies the effects of similar influence structure between the agents and local populations to [Narang et al., 2022, Piliouras and Yu, 2022].
> >
> > We would be happy to discuss with the reviewer should there be any other concerns/questions. We would kindly ask the reviewer to consider raising his/her score if your concerns are addressed. Thank you very much!
> >
> > **References**
> >
> > Sun, H., Lu, S. and Hong, M., Improving the sample and communication complexity for decentralized non-convex optimization: Joint gradient estimation and tracking. In ICML 2020.
> >
> > Shi, W., Ling, Q., Wu, G. and Yin, W.,. Extra: An exact first-order algorithm for decentralized consensus optimization. SIOPT, 2015.
> >
> > Roy, A., Balasubramanian, K. and Ghadimi, S. Projection-free Constrained Stochastic Nonconvex Optimization with State-dependent Markov Data. arXiv preprint arXiv:2206.11346, 2022.

---

### Official Review · Reviewer_hRQZ · 2022-07-11

**Rating:** 7
**Confidence:** 4
**Soundness:** 4 excellent
**Presentation:** 4 excellent
**Contribution:** 3 good

**Summary:**

The paper addresses multi-agent performative prediction (Multi-PfD), focusing on a setting where multiple agents are learning a common decision vector from data that can be influenced by their decisions. The paper formulates this as a decentralized optimization problem and identifies a necessary and sufficient condition for a unique multi-agent performative stable (Multi-PS) solution. The paper demonstrates that the consensus effect results in a relaxed requirement compared to the single agent case. As a solution to the problem, the paper studies a decentralized extension to a greedy deployment scheme (DSGD-GD), demonstrates that it converges to the Multi-PS solution, and analyzes the non-asymptotic convergence rate. The analysis reveals that heterogeneous users and poor graph connectivity may impair convergence but do not affect the asymptotic rate. Some numerical experiments on synthetic and real data are provided as validation of the analysis.

**Questions:**

1.	Is there a clear example where the consensus solution is clearly superior and desirable? i.e., Can the authors provide a stronger motivation for the studied problem?
2.	There is relatively limited discussion of the assumption that the local distribution is only affected by the agent’s decision vector. Can the authors elaborate on how this restricts the application of the results to multi-agent problems? Is it possible to extend the results to handle this setting?
3. As the authors acknowledge, the assumption of strong convexity is limiting. Can the authors comment on how the results might be extended to incorporate weaker requirements?


**Limitations:**

Yes.

**Strengths And Weaknesses:**

Strengths

1.	The paper studies an interesting problem and provides a thorough and useful theoretical analysis that addresses key questions of interest (existence of stable solution, convergence + rate). Although parts of the proofs are similar to those of related results in decentralized optimization and performative prediction, there are additional challenges in the analysis and these are handled in an elegant fashion.
2.	There is valuable discussion associated with the theoretical results. Going beyond expressions for bounds, the papers discusses key contributing factors to various terms. This leads to a specification of how factors such as graph connectivity impact the behaviour.
3.	The assumptions underpinning the analysis do not impose unreasonable constraints that make the setting impractical, although the strongly convex loss assumption and the local decision impact do impose limitations.
4.	The paper is very well written. The key theoretical results are presented in a clear manner and there is enough discussion of the proof procedure to provide the reader with an understanding of the general strategy.
5.	The numerical experiments, while limited and not the main focus of the paper, do provide a validation of the theoretical analysis.

Weaknesses

1.	There is a relatively weak motivation for the setting under study in the paper. In the introduction, the paper cites Hardt et al., Dong et al., and Kleinberg et al. in providing an example scenario, but the extension to the multi-agent setting is not clearly motivated. The example is a spam classifier and the proposed multi-agent context is regional servers with users that respond in different ways. In this case, it’s not obvious why the consensus solution is desirable. If there is sufficient data, then it would seem that individual classifiers for each region would be preferable. If not, then I would think that it would be natural to explore compromises with some parameters being shared and others permitting a suitable response to the local data.
2.	Experiments: While the experiments are primarily illustrative and serve the role of validating the theory, it is misleading and incorrect to claim that the paper “illustrate[s] the Multi-PfD problem in a real application” (l. 302). While the analyzed dataset is from a real application, neither the division into multiple agents nor the users’ proposed adaptation are.

---

> ### Author Response · Authors · 2022-08-02
> **Reply to Reviewer hRQZ - Part 1/2**
>
> We are glad that the reviewer liked our paper recognizing the theoretical contributions of our work to the Multi-PfD problem with cooperative agents, which is a new problem that holds promises to deepen our understanding of ML systems in real applications. Our point-to-point responses to concerns on weaknesses and questions are:
>
> > The example is a spam classifier and the proposed multi-agent context is regional servers with users that respond in different ways. In this case, it’s not obvious why the consensus solution is desirable. If there is sufficient data, then it would seem that individual classifiers for each region would be preferable.
>
> Thank you for the thoughtful comment. The consensus requirement is motivated by a common setup in decentralized learning where agents are willing to ***cooperate***. In the literature, this requirement is actually ubiquitous in decentralized learning [Lian et al., 2017], and is also relevant to federated learning where a common/similar model is sought, especially for applications to finance, AI as service [Kairouz et al., 2021], clinical data [Warnat-Herresthal et al., 2021]. In fact, the consensus design is desirable not only when each agent holds a small amount of data, but also when the agents hold heterogeneous data; see Fig. 1. Finding a common decision vector leads to a model to fit the data available at all agents, which is a desirable property for applications. Besides the above examples on decentralized learning, finance, etc., more concrete problems can be described. First example is the spam email filtering task in the paper, where the regional servers can be controlled by a single company who aims to design just one spam filtering product for the global market. Another example is when a network of banks are deciding on the loan/deposit interest rates, the latter needs to be set as the same. Note that the example also entails heterogeneous data as the interest rates can be learnt from customers going to the banks, who may reside in different regions with varying economic development levels.
>
> > If not, then I would think that it would be natural to explore compromises with some parameters being shared and others permitting a suitable response to the local data.
>
> We agree that partial consensus would be an interesting extension. Thank you for the suggestion! It is indeed reasonable to consider a partially compromised setup with some shared and private parameters, just like in personalized learning. However, we remark that such personalized design may lead to instability of the Multi-PfD problem as the latter is no longer in full cooperative mode. It is because the personalized problem contains a single agent learning component which cannot be aided by consensus anymore. Nevertheless, this is an interesting scenario worthy of further investigation.
>
> > It is misleading and incorrect to claim that the paper “illustrate[s] the Multi-PfD problem in a real application” (l. 302). While the analyzed dataset is from a real application, neither the division into multiple agents nor the users’ proposed adaptation are.
>
> We agree and thank the reviewer for pointing this out. Indeed, the said numerical experiment only “simulates the performance of Multi-PfD based on a real dataset”. We have corrected it in the revised paper.
>
> > Is there a clear example where the consensus solution is clearly superior and desirable? i.e., Can the authors provide a stronger motivation for the studied problem?
>
> Thank you for the question. An advantage of the consensus solution as proven lies in the improved stability condition under the effects of distribution shifts induced by the decision vectors; cf. Proposition 1. Admittedly it is a theoretical condition that can be difficult to directly demonstrate. We tried our best to illustrate it by the experiment in Fig. 1 on a Gaussian Mean Estimation problem. It is observed that under the same setup where a stable solution can be found by DSGD (with consensus solution), yet if an agent does not follow a consensus design, its decision vector will diverge.
>
> Moreover, we add that this problem in Fig. 1 pertains to practical problems involving quadratic cost/utility functions that are commonly studied in the game theory literature; see Sec 7.2, [Narang et al., 2022].

---

> > ### Author Response · Authors · 2022-08-02
> > **Reply to Reviewer hRQZ - Part 2/2**
> >
> > > There is relatively limited discussion of the assumption that the local distribution is only affected by the agent’s decision vector. Can the authors elaborate on how this restricts the application of the results to multi-agent problems? Is it possible to extend the results to handle this setting?
> >
> > Thank you for the question. This assumption has been directly motivated by our targeted application of decentralized learning when the users interacting with different agents are naturally separated, e.g., through geographical means. This is a common scenario for applications in medical and economic fields. Due to the geographical barriers, the outcomes/data from these users tend not to be influenced by the decisions of other agents. We believe that this is not a restriction of our model, but an accurate modeling of the real world.
> >
> > In fact, we can also relax the assumption to ${\cal D}_i$ that depends on $\theta_1,...,\theta_n$ similar to [Narang et al., 2022]. This can be done by extending A3 and modifying the proof of Lemma 3. We have included a new section, Appendix H, in the revised paper to discuss this issue.
> >
> > > As the authors acknowledge, the assumption of strong convexity is limiting. Can the authors comment on how the results might be extended to incorporate weaker requirements?
> >
> > Thank you for the question. To shed some light, we note that only a few recent works have partial results on the matter [Li and Wai, 2022], [Roy et al., 2022] for single agent performative prediction, where the convergence to a near-stationary point is established. For instance, [Li and Wai, 2022] achieves this result by treating the greedy-deployment scheme (which is akin to DSGD-GD) as a biased SA algorithm. For Multi-PfD, we anticipate that a similar proof will work, but it will also stop at yielding the convergence to a near-stationary point, which is a weaker form of convergence than in our Theorem 1.
> >
> > We would be happy to discuss with the reviewer should there be any other concerns/questions. Thank you very much!
> >
> > **References**
> >
> > S. Warnat-Herresthal et al., Swarm learning for decentralized and confidential clinical machine learning, Nature, 2021.
> >
> > P. Kairouz et al., Advances and Open Problems in Federated Learning, Foundation and Trends in Machine Learning, 2021.
> >
> > Roy, A., Balasubramanian, K. and Ghadimi, S., Projection-free Constrained Stochastic Nonconvex Optimization with State-dependent Markov Data. arXiv preprint arXiv:2206.11346, 2022.

---

> > > ### Comment · Reviewer_hRQZ · 2022-08-07
> > > **Rebuttal acknowledgement**
> > >
> > > Thank you for the response to my concerns/questions and those of the other reviewers. I appreciate the modification of the text concerning the analysis of the real-world dataset. I think Appendix H adds useful material and shows that results are not reliant on the local impact assumption. The response to the question about the strong convexity assumption seems reasonable to me, and I don't think it is a major limitation of the current work.
> > >
> > > While I have some lingering doubts about the practicality of the example problem setting, the discussion in the response is reasonable. The provided examples (e.g., clinical data) primarily pertain to decentralized or federated learning, and there isn't a clear connection to the performative prediction context. But given that this is a relatively new field of study, and the performative aspect is often related to behaviour responses to a deployed system, it is likely that more practical examples will emerge over time.
> > >
> > > I have carefully read the other reviews and the authors' responses. One of the other reviewers also questioned motivation, and I will be interested to see if the response satisfactorily addressed the concern. I don't see that the other reviewers identified other significant weaknesses in the paper. As I stated in my original review, I think the paper does make a valuable, novel theoretical contribution and the problem is not overly artificial (a compelling practical example would serve to strengthen the paper but I don't think the absence undermines it). With this in mind, I am retaining my initial recommendation to accept the paper.

---

> > > > ### Author Response · Authors · 2022-08-08
> > > > **Reply to Reviewer hRQZ**
> > > >
> > > > We thank reviewer hRQZ for the positive comment on our paper.
> > > >
> > > > For your concerns about the connection between the provided examples and performative prediction, we would like to point out that the effects of performative prediction are ubiquitous in applications involving outcomes that are controllable by the environment that can be a different party than the agents. As an example, in clinical data, the agents wish to train a model for treating some medical conditions. Upon the deployment of a model, the future patients may react to the just deployed model by overstating their symptoms accordingly, which increase their chances of receiving better treatment at the hospital.

---

### Meta-Review · Area_Chair_Lxpi · 2022-08-24

**Recommendation:** Accept
**Confidence:** Certain

**Metareview:**

This paper received a mixed set of reviews. After reading the paper, the reviews, subsequent author-reviewer discussion, and discussing with the reviewers, I recommend that this paper be accepted. The paper studies a multi-agent version of the performative prediction problem. The paper contains original and interesting contributions, and we expect the community to appreciate this work. In particular, the theoretical contribution was appreciated by several reviewers. Although concerns were raised during the review process, my impression is that these were satisfactorily addressed through the rebuttals.

While preparing the camera ready, we strongly encourage the authors to address the main concerns that came up in reviews. These include:
* Strengthening the motivation for the particular multi-agent formulation studied in this paper
* Clarifying the relationship between performative prediction and prior work on distributed optimization

**Award:**

No

---

### Decision · Program_Chairs · 2022-09-14

Accept